# A Fuzzy Prescriptive Analytics Approach to Power Generation Capacity Planning

Berna Tektaş [1], Hasan Hüseyin Turan [2], Nihat Kasap [3], Ferhan Çebi [4] and Dursun Delen [5,6,*]

1    Faculty of Business Administration, Izmir Katip Celebi University, Cigli, Izmir 35620, Turkey;
berna.tektas@ikcu.edu.tr
2    School of Engineering and Information Technology, University of New South Wales,
Canberra, ACT 2612, Australia; h.turan@adfa.edu.au
3    Sabanci Business School, Sabanci University, Tuzla, Istanbul 34956, Turkey; nihatk@sabanciuniv.edu
4    Management Faculty, Istanbul Technical University, Maçka, Istanbul 34367, Turkey; cebife@itu.edu.tr
5    Spears School of Business, Oklahoma State University, Stillwater, OK 74078, USA
6    Faculty of Engineering and Natural Sciences, Istinye University, Istanbul 34396, Turkey
*    Correspondence: dursun.delen@okstate.edu or dursun.delen@istinye.edu.tr; Tel.: +1-(918)-850-0709;
Fax: +1-(918)-594-8281

**Abstract:** This study examines the long-term energy capacity investment problem of a power generation company (GenCo), considering the drought threat posed by climate change in hydropower resources in Turkey. The mid-term planning decisions such as maintenance and refurbishment scheduling of power plants are also considered in the studied investment planning problem. In the modeled electricity market, it is assumed that GenCos conduct business in uncertain market conditions with both bilateral contracts (BIC) and day-ahead market (DAM) transactions. The problem is modeled as a fuzzy mixed-integer linear programming model with a fuzzy objective and fuzzy constraints to handle the imprecisions regarding both the electricity market (e.g., prices) and environmental factors (e.g., hydroelectric output due to drought). Bellman and Zadeh's max-min criteria are used to transform the fuzzy capacity investment model into a model with a crisp objective and constraints. The applicability of methodology is illustrated by a case study on the Turkish electric market in which GenCo tries to find the optimal power generation investment portfolio that contains five various generation technologies alternatives, namely, hydropower, wind, conventional and advanced combined-cycle natural gas, and steam (lignite) turbines. The results show that wind turbines with low marginal costs and steam turbines with high energy conversion efficiency are preferable, compared with hydroelectric power plant investments when the fuzziness in hydroelectric output exists (i.e., the expectation of increasing drought conditions as a result of climate change). Furthermore, the results indicate that the gas turbine investments were found to be the least preferable due to high gas prices in all scenarios.

**Keywords:** generation investment planning; maintenance and refurbishment scheduling; uncertainty; fuzzy mathematical programming; climate change

## 1. Introduction

A good strategic planning process that considers the effects of climate change and changes in the economic environment for the growth of the electric power system is critical to meeting the rapidly increasing demand for electricity. The Generation Expansion Planning problem (GEP) typically deliberates a 10–20-year planning horizon. GEP problems deal with scheduling the changes (acquisitions of expansions) on capacity, deciding the technology to invest in, and the timing of new power plant investments to meet the projected load demand.

The changes in the economic environment (e.g., prices and cost changes) and the electric power generation industry (e.g., new technological developments) are likely to

affect the investment planning decisions of the generation companies (GenCos). These changes usually result in fluctuations (i.e., demand uncertainties) in demand in the electricity market. Neglecting uncertainties and using deterministic models would lead to suboptimal investment decisions for a GenCo. It is essential to develop more realistic models to incorporate randomness. In this direction, we model GenCo's objective (i.e., minimization of total cost) and constraints by fuzzy logic methods.

There are two approaches, centralized and decentralized, related to GEP studies in the literature [1]. In the centralized approach, the GEP exercise is generally undertaken by the central planner. Thus, the government-owned or private-utility monopolies solve the GEP problem centrally, in order to ensure a system reliability level while meeting the power demand growth at the minimum system-wide plan costs. Additionally, the GEP problem is solved centrally by the governing or regulating authorities in deregulated electricity markets, to formulate market designs and policies that lead to the long-term targets of a country concerning the minimization of the overall cost of supplying electricity to the users, the penetration of renewable energy technologies, renewable energy sources support schemes and/or the green-house gasses emission control [1].

There is a substantial amount of study in the literature that uses the centralized approach to solve the GEP problem. Several methods such as linear programming [2], mixed-integer [3] mixed-integer linear programming [4], dynamic programming [5], non-linear programming [6] multi-objective programming [7], stochastic programming [8] fuzzy logic (fuzzy programming) [9], and meta-heuristic approaches such as swarm optimization [10] and the Evolution Algorithm [11] are used to decide the best capacity, technology type and investment timeline of generation unit to sustain energy demand. The formulation of the objective function and constraints of the (these) GEP problem(s) varies, such as incorporating emissions costs and other environmental constraints, reliability criteria, reserve margins, demand-side management programs, transmission constraints, financial and location constraints [12,13]

In the decentralized approach, the GEP problem is considered a strategic decision-making problem of a GenCo operating in a deregulated electricity market. After the California electricity crisis, also known as the Western US Energy Crisis of 2000 and 2001, researchers started to focus on solving challenging and complex decentralized GEP problems that have modeled the electricity market operations and the behaviors of the rival GenCos together with the investment decisions of the strategic GenCo (http://www.eia.gov/electricity/policies/legislation/california/subsequentevents.html, accessed on 1 December 2021). Game theory is engaged for almost all similar models, solved by stochastic dynamic programming [14], Lagrangian relaxation and Benders decomposition [15,16] evolutionary programming [17] Genetic Algorithm [18], system dynamics [19], and swarm optimization [20,21] Each of these methods has its advantages and disadvantages; a comparative analysis of these methods can be found in [22].

### 1.1. Climate Change

One of the uncertainties is related to environmental issues as consequences of future climate changes. Recent studies show that the frequency, severity, and duration of excessive weather conditions such as drought are expected to increase in the future due to climate change [23]. For example, Turkey is situated in the Mediterranean region, and hence, it is expected that Turkey's water sources will be affected by global climate change [24]. Using a simulation-based study, Aktas [24] demonstrated that nearly 20% of the surface water in the studied basins in Turkey will be vanished by the year 2030, and the trend for this loss in water reserve will increase to 35% in 2050 and 50% in 2100. Furthermore, the same study also suggests that the evapotranspiration of plants will contribute to the water loss by as much as another 20%. The second source of uncertainty arises due to financial and budgeting issues. To illustrate, GenCo may tolerate some increase in the planned annual investment budget by using the owned current credit facilities and/or via equity capital.

This study examines a 10-year investment-planning problem of a liberated GenCo. GenCo sells the generated power via bilateral contracts (BICs) and the day-ahead market (DAM). Electricity market reports of the Republic of Turkey Energy Market Regulatory Authority (EMRA) show that Turkey has a high demand growth rate [25]. Therefore, it is assumed that GenCo can sell all its generated electricity to the electricity market, thereby implying that there is enough demand in the electricity system to absorb all the energy generated by GenCo. The GenCo may invest in five various generation technologies: wind, hydropower, steam (lignite) turbines, and conventional and advanced combined-cycle natural gas (CCCNG and ACCNG), as depicted in Figure 1. Additionally, we assumed that adequate transmission resources are in place to transfer the power generated by the new units to the demand locations. This assumption implies that the GenCos investment decisions are not affected by transmission constraints. Additionally, it is assumed that the investment decisions of the investors do not influence the electricity market prices. This assumption ensures that no GenCo holds market power or is likely to hold market power through large-scale investments. In our model, it is assumed that GenCo is the sole owner and operator of the plants. This assumption means that the dispatch decisions are taken by the firm.

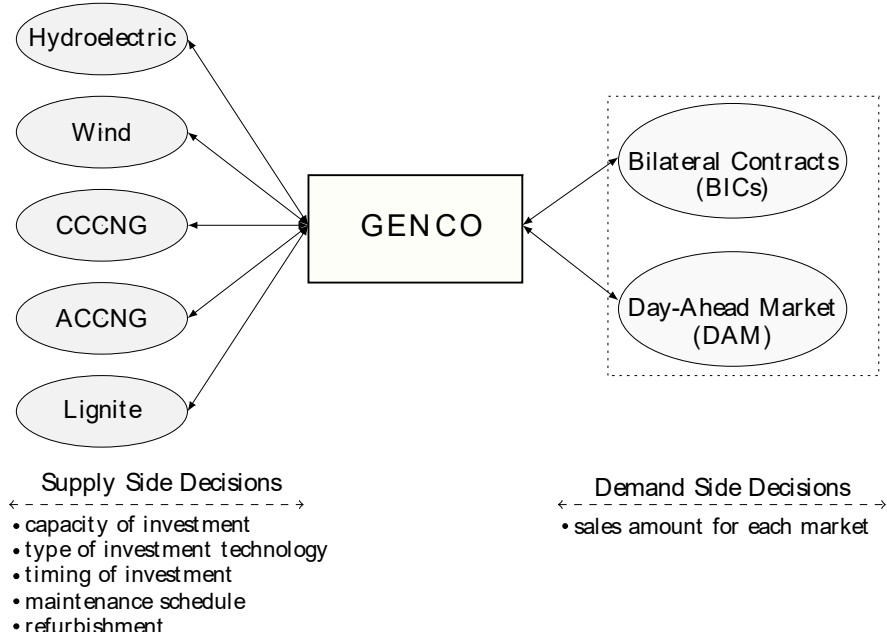

**Figure 1.** Electricity market environment for a GenCo.

There are several studies in the literature show that the global climate change has adverse impacts both on the financial performance of both existing and potential power plant investments according to their locations and on the availability of the hydroelectric resources (the hydroelectric potential of specific basins) [24,26–28]. It was shown in [29] that global hydroelectric potential due to global climate change will be affected very little; however, the hydroelectric potential of some countries increases, whereas some others are expected to lose theirs. They also show that, in both cases, risk levels such as potential flooding and drought are high.

This manuscript investigates which type of power plant investments can be substituted with hydropower investments in the long run against drought risks due to climate change, and which types of power plants meet the decrease in hydroelectric production because of drought. There is almost no study in the literature about what kind of electrical energy generation technology investments can be used to substitute the hydroelectric generation unit investments in case of drought. In their study, ref. [30] show that due to the slow work-pace of the coal supply chain, the losses in hydroelectric energy output during drought

periods in Turkey can initially be compensated by natural gas power plants. However, later, the coal power plants increase their production. They included the utilization dimension of power plants; however, they did not consider the effects of drought expectation in investment plans as we included in this manuscript.

Reference [31] examines approaches from various perspectives and presents a renewed and complete survey of the optimization method implementation for the hydro scheduling solution. Similarly, it was presented by [32] a comprehensive state-of-the-art survey on power generation expansion planning with renewable energy sources. According to [33], there are several ways by which electric power infrastructure has contributed to climate change, and how climate change affects electric power infrastructure. The optimal Generation System Expansion Plan was studied by [34] that can satisfy the increasing electricity demand while maintaining operational elements and the stability of the energy supply. They included the maintenance dimension; however, they did not consider environmental factors as we included in this manuscript.

### 1.2. Fuzzy Logic Modeling

The developed long-term (strategic) investment-planning model considers the fuzziness in the hydroelectric output and annual budget. The developed fuzzy mixed-integer linear programming model is solved by using the Zimmermann approach [35,36]. Afterward, a scenario-based (sensitivity) analysis is performed to investigate how the investment decisions of the GenCo vary when drought expectation and fuzziness (uncertainty) in the maximum energy output of hydroelectric units increase. In the proposed model, the location effect of generation investment decisions is also incorporated. Especially, wind turbine and hydroelectric generation unit capacities highly depend on both locational and climatic behaviors of the selected region. Conversely, the lignite power plants' operational cost is affected by the location since the distance of the coal mine (or a harbor that could be a hub for coal supply) is an important factor for their operational costs.

Fuzzy logic enables uncertain information or elusive information to be managed; for many years, it has been used in the literature as an effective method for solving problems and modeling systems that contained uncertain and indefinite information [30]. Ever since 1990, fuzzy logic has found its place in various practices regarding energy management and planning [37,38]. Regarding electricity generation investment planning, fuzzy set theory and fuzzy logic have been used for multi-purpose planning and identifying uncertainties or qualitative requirements in electrical power generation expansion planning, along with techniques such as linear programming, mixed-integer linear programming, or dynamic programming [39].

One of the peculiar aspects of the model mentioned above is its inclusion of fuzziness in a decision environment, along with medium-term sales and care planning that may exert influence on the actions and investment decisions of GenCo in two different wholesale markets. Another peculiar aspect of the study is its evaluation of the effect brought on by the tolerance (deviation ratio) regarding the maximum electrical energy output of hydroelectric generation units for GenCo's optimal investment decisions via using scenario analyses.

Turkey is placed in the Mediterranean region which will have a negative impact on its water sources from global climate change [24]. Even though the generation of hydroelectric is lessened in dry spells, it does not go past the maximum turbine flow in times of heavy precipitation. In other words, the generation of hydroelectric is very sensitive to short- and long-term climatic variabilities [40]. The sensitivity analyses were carried out to examine the substitution between the electrical energy power generation sources as the fuzziness increases regarding the losses experienced in the electricity generation of the hydroelectric power units.

As a contribution of this study, the proposed model considers the operations of the GenCo in the BIC market and DAM, in an integrated manner by using short (monthly) periods for long-termed (strategic) GEP modeling. Thus, both the mid-termed sale and

power generation strategy of GenCo are deliberated in the strategic investment planning. Moreover, mid-term decisions such as maintenance scheduling of the generation units are considered in a decentralized long-termed GEP model. Additionally, different from [1,22,41], this study used monthly equivalent forced outage rate modeling (EFOR) for each planning year in order to consider seasonal changes in the availability of generation units, especially units using renewable energy sources. However, not only the EFOR rate but also the precipitation and evaporation level will affect the amount of optimistic and pessimistic hydropower output. That is why the proposed model considers not only the technological limits of hydroelectric power plants, but also fuzziness in their water resources caused by drought. Using short periods in long-termed GEP modeling helps to investigate both generation amount changes of power plants to compensate for the decrease in hydropower generation during drought, and how investment plans change according to the drought expectation.

The remainder of the paper is structured in the following manner. Section 2 presents the mathematical formulation of the fuzzy investment model and depicts the solution algorithm. Section 3 depicts the case study of the proposed model. Section 4 delivers a thorough examination of the case study findings. Finally, Section 5 provides concluding remarks and future research directions.

## 2. The Proposed Model Considering MTP for GenCos' Investment Planning

Due to economic uncertainties, environmental conditions that affect the electricity generation of power units using renewable energy resources, and deviations from the projected investment budgets, decisions concerning electrical energy power generation are made in a fuzzy environment with incomplete and inaccurate information. Hence, in this study, an investment planning model that considers medium-term planning (MTP), is formulated. The model contains a fuzzy objective function, an annual investment budget, and the output of electricity from hydroelectric energy plants, which is deemed to be the crucial component of the currently installed power of the GenCo. Consideration of monthly sub-periods along with strategic investment planning decisions reduce planning risks and also affect the present value of the total profit of GenCos. This approach also enables medium-term evaluation (monthly for each planning year), maintenance scheduling, and sales planning decisions, together with long-term investment planning decisions. Moreover, medium-term planning considers seasonal and climate changes, especially for the units using renewable energy sources.

### 2.1. Mathematical Formulation

The notation is given in the Abbreviations. The fuzzy generation expansion planning is given in Equations (1) thru (24). The objective function in Equation (1) shows the maximization of the agreed satisfaction level ($\lambda \in [0,1]$) of the fuzzy goal, the GenCo's total profit target in constraint (2), and two fuzzy constraints, (3) and (4), and the energy production constraint of hydroelectric units and annual budget constraint for new capacity additions.

$$\text{Maximum } \lambda \tag{1}$$

$$
\begin{aligned}
&\sum_y DF_y \sum_m \sum_b D_{y,m,b} \left( Pr_{y,m,b}^{\text{BIC}} p_{y,m,b}^{\text{BIC}} + Pr_{y,m,b}^{\text{DAM}} p_{y,m,b}^{\text{DAM}} \right) \\
&- \sum_y DF_y \sum_m \sum_b D_{y,m,b} \sum_i C_{i,y}^{\text{VOM}} p_{i,y,m,b} - \sum_y DF_y \sum_i C_{i,y}^{\text{FOM}} P_{i,y}^{\text{Max}} u_{i,y} \\
&- \sum_y DF_y \sum_{i \in I^Y} P_{i,y}^{\text{Max}} C_i^{\text{Inv}} \omega_{i,y} - \sum_y DF_y \sum_{i \in I^R} P_{i,y}^{\text{Max}} C_i^R x_{i,y} \\
&+ DF_Y \sum_{i \in I^C} P_{i,y}^{\text{Max}} C_i^{\text{Inv}} \sum_y \left( \omega_{i,y} - \frac{1}{T_i^L} u_{i,y} \right) + DF_Y \sum_{i \in I^R} P_{i,y}^{\text{Max}} C_i^R \sum_y \left( x_{i,y} - \frac{1}{T_i^{\text{L-R}} - T_i^L} v_{i,y} \right) \\
&\geq Z^- + (Z^+ - Z^-)\lambda
\end{aligned}
\tag{2}
$$

$$\sum_m \sum_b D_{y,m,b} p_{i,y,m,b} + \left( \pi_{O_{i,y}^H}^+ - \pi_{O_{i,y}^H}^- \right) \lambda \leq \pi_{O_{i,y}^H}^+ \quad \forall i \in I^H, \forall y \in Y \tag{3}$$

$$\sum_{i \in I^{C}} P_{i,y}^{\text{Max}} C_i^{\text{Inv}} \omega_{i,y} + \sum_{i \in I^{R}} P_{i,y}^{\text{Max}} C_i^{\text{R}} x_{i,y} + \left( \pi_{Bud_y}^{+} - \pi_{Bud_y}^{-} \right) \lambda \leq \pi_{Bud_y}^{+} \quad \forall y \in Y \tag{4}$$

$$p_{i,y,m,b} \leq \left( 1 - EFOR_{i,y,m} \right) P_{i,y}^{\text{Max}} u_{i,y} \quad \forall i \in I, \forall y \in Y, \forall m \in M, \forall b \in B \tag{5}$$

The left-hand side of Equation (2) has seven terms. Total revenue (*TR*) received by power sales via BICs and spot market given in the first term. The total variable operation and maintenance (O&M) cost (*TVOC*) and total fixed O&M cost (*TFOC*) of all generation units, new generation units' total investment cost (*TIC*), and total refurbishment cost of existing units (*TRC*) given in the second, third, fourth and fifth terms, respectively. Finally, in the sixth and seventh terms, the present value of the salvage worth of the new generation units (*SV*) and the salvage value of the refurbished units (*RSV*) are added to the fuzzy profit function of the GenCo. Hence, the left-hand side of Equation (2) can be written as

$$Z = TR - TVOC - TFOC - TIC - TRC + SV + RSV$$

The pressure and the flow rate of water that turn the turbine is the main factor in determining the turbine type used in a hydroelectric power plant. There are three types of turbines used in today's hydroelectric power plants: Kaplan, Francis, and Pelton turbines. Constraint (3) shows that the maximum hydroelectric output of a power plant is limited because of the chosen turbine technology. However, not only the selected turbine technology but also the precipitation and evaporation level will affect the amount of optimistic and pessimistic hydropower output. That is why Constraint (3) is set to consider not only the technological limits of hydroelectric power plants, but also fuzziness in their water resources caused by drought. Fuzzy constraint (4) ensures that the cost for new capacity addition in a particular year, y has to be between the optimistic and pessimistic values of the annual budget allocated for year y. This constraint implies that GenCo may tolerate an increase in the annual investment budget by using owned credit facilities if it is needed. The linear membership function of the annual budget is expressed with Equation (30) and shown in Figure 4. Constraint (5) is set to consider not only the technological unavailability of generation units but also their resource unavailability. EFOR modeling in Equation (5) enables decision-makers to consider partial or complete unplanned outages of generation units, especially of wind turbines, that are caused by natural conditions. Monthly EFOR modeling is used for each planning year in order to consider seasonal changes.

$$\sum_{i} p_{i,y,m,b} - \left( p_{y,m,b}^{\text{BIC}} + p_{y,m,b}^{\text{DAM}} \right) = 0 \quad \forall i \in I, \forall y \in Y, \forall m \in M, \forall b \in B \tag{6}$$

$$p_{y,m,b}^{\text{BIC}} \geq \alpha_y \sum_{i} p_{i,y,m,b} \quad \forall y \in Y, \forall m \in M, \forall b \in B \tag{7}$$

$$p_{y,m,b}^{\text{BIC}} \leq \beta_y \sum_{i} p_{i,y,m,b} \quad \forall y \in Y, \forall m \in M, \forall b \in B \tag{8}$$

$$\sum_{i} P_{i,y}^{\text{Max}} u_{i,y} \leq \theta P_{y-1}^{\text{TR}} \quad \forall y \in Y \tag{9}$$

The problem constraints associated with the electricity wholesale market are given in Equations (6)–(9). Short cyclical changes in electricity prices can affect GenCos' medium-term generation plans and, therefore, their profitability and long-term investment decisions. Monthly bilateral contract and day-ahead market prices are considered for each planning year and develop constraint Equations (6)–(8) with monthly sub-periods. The balance between generated power output and the power sold by the GenCo is represented in Equation (6). With Equation (7), the GenCo hedges the volatility risk in DAM prices by specifying a lower required amount for its BICs, and also with Equation (8) GenCo specifies

an upper limit for BICs for not missing high-profit opportunities in DAM. The competition protection rule of the Electricity Market is included in the GEP model by Equation (9).

$$\sum_m pm_{i,y,m} = 1 \quad \forall i \in I, \forall y \in Y \tag{10}$$

$$p_{i,y,m,b} \le P_{i,y}^{\text{Max}} \left(1 - pm_{i,y,m,b}\right) \quad \forall i \in I, \forall y \in Y, \forall m \in M, \forall b \in B \tag{11}$$

$$pm_{i,y,m} + \sum_{i' \ne i} pm_{i,y,m} \le 1 \quad \forall i, i' \in s, \forall s \in S, \forall y \in Y, \forall m \in M \tag{12}$$

Equations (10)–(12) are constraints on the maintenance scheduling of generation units. The assumption that each generation unit is taken out of service once a year for one month is included in the model by Equation (10). Equation (11) ensures that if a generation unit is taken out of service for maintenance, it also cannot produce power during its maintenance period (1 month *i* each year). Equation (12) forces the generation units in the same power plant to be under maintenance in different months of the year for avoiding disabling the power plant completely.

$$\sum_y \omega_{i,y} \le 1 \quad \forall i \in I^C, \forall y \in Y \tag{13}$$

$$\sum_{\tau = y - T_i^L - 1}^{y} \omega_{i,\tau} = u_{i,y} \quad \forall i \in I^C, \forall y \in Y \tag{14}$$

$$u_{i,y} = 0 \quad \text{if } y \le T_i^{\text{Cons}} \quad \forall i \in I^C \tag{15}$$

$$u_{i,y} = \left\{ \begin{array}{l} -T_i^{\text{Ini}} \le y \le T_i^L - T_i^{\text{Ini}}, 1 \\ \text{otherwise}, 0 \end{array} \right. \quad \forall i \in I^E \text{ and } i \notin I^R \tag{16}$$

$$\omega_{i,y} = \left\{ \begin{array}{l} y = T_i^L - T_i^{\text{Ini}}, 1 \\ \text{otherwise}, 0 \end{array} \right. \quad \forall i \in I^E \tag{17}$$

$$u_{i,y} = \left\{ \begin{array}{l} -T_i^{\text{Ini}} \le y \le T_i^L - T_i^{\text{Ini}}, 1 \\ y > T_i^{L_R} - T_i^{\text{Ini}}, 0 \end{array} \right. \quad \forall i \in I^R \tag{18}$$

$$u_{i,y} = v_{i,y} \quad \forall i \in I^R, y > T_i^L - T_i^{\text{Ini}} \tag{19}$$

$$x_{i,y} = 0 \text{ if } y \le \left(T_i^L - T_i^{\text{Ini}}\right) \text{ and } y > \left(T_i^L - T_i^{\text{Ini}} + 1\right) \quad \forall i \in I^R \tag{20}$$

$$v_{i,y} = 0 \text{ if } y \le \left(T_i^L - T_i^{\text{Ini}}\right) \quad \forall i \in I^R \tag{21}$$

$$x_{i,y} = v_{i,y} \text{ if } y = T_i^L - T_i^{\text{Ini}} + 1 \quad \forall i \in I^R \tag{22}$$

$$v_{i,y} = v_{i,y-1} \text{ if } y > T_i^L - T_i^{\text{Ini}} + 1 \text{ and } y \le T_i^{L_R} - T_i^{\text{Ini}} \quad \forall i \in I^R \tag{23}$$

$$0 \le \lambda \le 1 \tag{24}$$

The refurbishment and start-up decisions and commissioning status of generation units are constructed by using binary constraints such as (13)–(23). Equation (13) states that the GenCo can finance a candidate unit once, Equation (14) restricts the lifetimes of a candidate unit to $T_i^L$ years, and also Equation (15) refers to that candidate units cannot be

commissioned before their construction time. It is assumed that the investment cost occurs with the related unit's commissioning (start-up) decision. Therefore, the existing units' commissioning status not eligible for refurbishment is given as input data by Equation (16), and the execution times of commissioning decisions for existing units are given as input data by Equation (17). Furthermore, Equations (18) and (19) define the existing units' commissioning status under refurbishment. It is assumed that the refurbishment decision for an existing generation unit can only be executed immediately after the end of its economic lifetime by Equation (20), and Equations (21) and (22) define the refurbished unit's commissioning status. In addition, Equation (23) limits the lifetime addition of a refurbished unit to $\left( T_i^{\text{L-R}} - T_i^{\text{L}} \right)$ years.

The fuzzy version of the objective function (Z) can generally be expressed with Equation (25).

$$\text{Maximum } \widetilde{Z} = TR - TVOC - TFOC - TIC - TRC + SV + RSV \tag{25}$$

Figure 2 provides the linear membership function of Equation (26).

$$\mu_Z = \begin{cases} 0 & , TR - TVOC - TFOC - TIC - TRC \\ & \quad +SV + RSV < Z^- \\ 1 - \dfrac{Z^+ - TR - TVOC - TFOC - TIC - TRC + SV + RSV}{Z^+ - Z^-} & , Z^- \leq TR - TVOC - TFOC - TIC \\ & \quad -TRC + SV + RSV \leq Z^+ \\ 1 & , TR - TVOC - TFOC - TIC - TRC \\ & \quad +SV + RSV > Z^+ \end{cases} \tag{26}$$

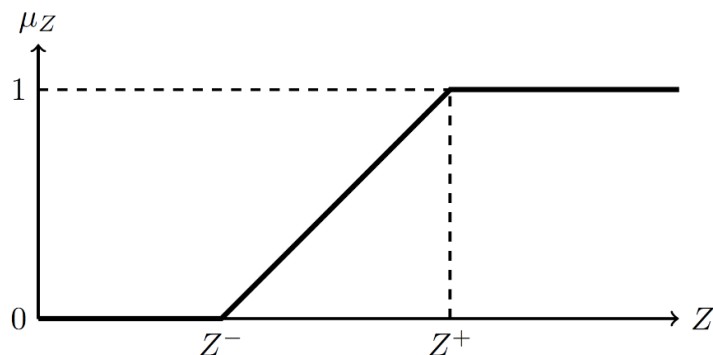

**Figure 2.** Fuzzy membership function for the objective function.

To find the element with the highest membership degree, which can satisfy both the fuzzy objective and fuzzy constraints simultaneously, an extra $\lambda \in [0,1]$ variable was defined. In the equivalent model that was devised by using the maximum-minimum operator of [42], the objective of the fuzzy investment planning model that takes account of MTP is the now highest value of $\lambda$ as shown in Equation (1), and the fuzzy objective is placed in the new model as a constraint similar to that of Equation (2)'s.

### 2.2. Fuzzy to Crisp Constraint Conversion

The electricity output of hydroelectric generation units in our model is limited to the maximum (optimistic) electricity output $\left( \pi_{O_{i,y}^H}^+ \right)$, which is the total of the absolute and secondary energy generation outputs calculated for the full upstream development status [1]. However, in practice, the electricity output of a given hydroelectric unit will generally be lower than the maximum electricity output of that hydroelectric unit. If the working hours of hydroelectric plants are lessened by 12 h each month, approximately a 5% drop will be seen in its yearly electricity output [30]. Any deviations that may

arise in maximum (optimistic) electricity output ($\pi^{+}_{O^{H}_{i,y}}$) during dry spells will affect the profitability and investment decisions of GenCo. The minimum possible electricity output level against the decreasing precipitation (climatic changes) is given as $\pi^{-}_{O^{H}_{i,y}}$ in our model.

The membership function of the constraint concerning the electrical generation output of hydroelectric plants is given in Equation (27).

$$
\mu_{O^{H}_{i,y}} = \begin{cases} 0 & , \sum_{m}\sum_{b} D_{y,m,b}p_{i,y,m,b} \geq \pi^{+}_{O^{H}_{i,y}} \\ 1 - \dfrac{\sum_{b} D_{y,m,b}p_{i,y,m,b} - \pi^{-}_{O^{H}_{i,y}}}{\pi^{+}_{O^{H}_{i,y}} - \pi^{-}_{O^{H}_{i,y}}} & , \pi^{-}_{O^{H}_{i,y}} \leq \sum_{m}\sum_{b} D_{y,m,b}p_{i,y,m,b} \leq \pi^{+}_{O^{H}_{i,y}} \\ 1 & , \sum_{m}\sum_{b} D_{y,m,b}p_{i,y,m,b} \leq \pi^{-}_{O^{H}_{i,y}} \end{cases} \tag{27}
$$

Hence, the fuzzy membership function of Equation (27) is given in Figure 3.

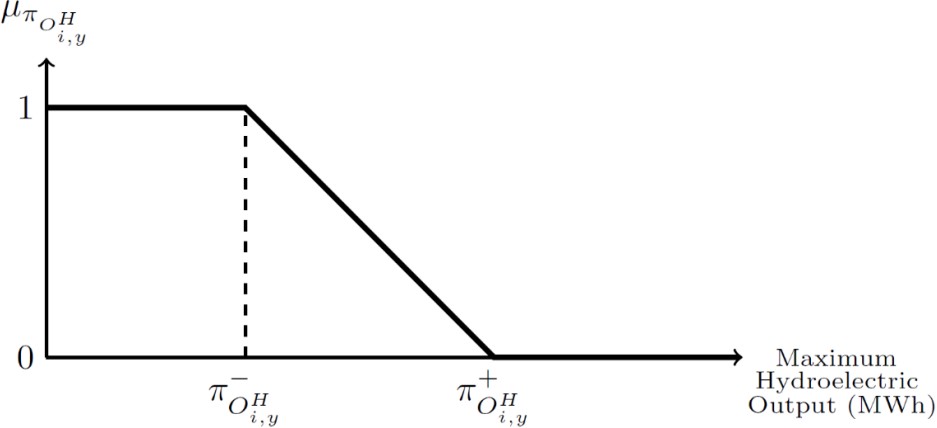

**Figure 3.** Hydropower generation constraint's fuzzy membership function.

The fuzzy constraint expressed in Equation (27) can be expressed with Equation (28) below for the fuzzy constraint regarding the electricity generation output of hydroelectric plants. When the necessary arrangements are made in Equation (28), Equation (3) will be obtained.

$$
1 - \frac{\sum_{m}\sum_{b} D_{y,m,b}p_{i,y,m,b} - \pi^{-}_{O^{H}_{i,y}}}{\pi^{+}_{O^{H}_{i,y}} - \pi^{-}_{O^{H}_{i,y}}} \geq \lambda \quad \forall i \in I^{H} \text{ ve } \forall y \in Y \tag{28}
$$

If the GenCo requires, it may tolerate an increase in the annual investment budget by using owned credit facilities. The optimistic and pessimistic values for the maximum budget are given as $\pi^{+}_{Bud_{y}}$ and $\pi^{-}_{Bud_{y}}$ in our model, respectively. The linear membership function of a fuzzy budget is expressed with Equation (29) and shown in Figure 4. The fuzzy constraint expressed in Equation (29) can be expressed with Equation (30) below for the fuzzy constraint regarding the annual investment budget. When the necessary arrangements are made in Equation (30), Equation (4) will be obtained.

$$
\mu_{Bud_{y}} = \begin{cases} 0 & , \sum_{i \in I^{A}} P^{\text{Max}}_{i,y}C^{\text{Inv}}_{i}\omega_{i,y} + \sum_{i \in I^{R}} P^{\text{Max}}_{i,y}C^{R}_{i}x_{i,y} > \pi^{+}_{Bud_{y}} \\ 1 - \dfrac{\sum_{i \in I^{A}} P^{\text{Max}}_{i,y}C^{\text{Inv}}_{i}\omega_{i,y} + \sum_{i \in I^{R}} P^{\text{Max}}_{i,y}C^{R}_{i}x_{i,y} - \pi^{-}_{Bud_{y}}}{\pi^{+}_{Bud_{y}} - \pi^{-}_{Bud_{y}}} & , \pi^{-}_{Bud_{y}} \leq \sum_{i \in I^{A}} P^{\text{Max}}_{i,y}C^{\text{Inv}}_{i}\omega_{i,y} + \sum_{i \in I^{R}} P^{\text{Max}}_{i,y}C^{R}_{i}x_{i,y} \leq \pi^{+}_{Bud_{y}} \\ 1 & , \sum_{i \in I^{A}} P^{\text{Max}}_{i,y}C^{\text{Inv}}_{i}\omega_{i,y} + \sum_{i \in I^{R}} P^{\text{Max}}_{i,y}C^{R}_{i}x_{i,y} < \pi^{-}_{Bud_{y}} \end{cases} \tag{29}
$$

$$1 - \frac{\sum\limits_{i \in I^A} P_{i,y}^{\text{Max}} C_i^{\text{Inv}} \omega_{i,y} + \sum\limits_{i \in I^R} P_{i,y}^{\text{Max}} C_i^R x_{i,y} - \pi_{Bud_y}^-}{\pi_{Bud_y}^+ - \pi_{Bud_y}^-} \geq \lambda \quad \forall y \in Y \qquad (30)$$

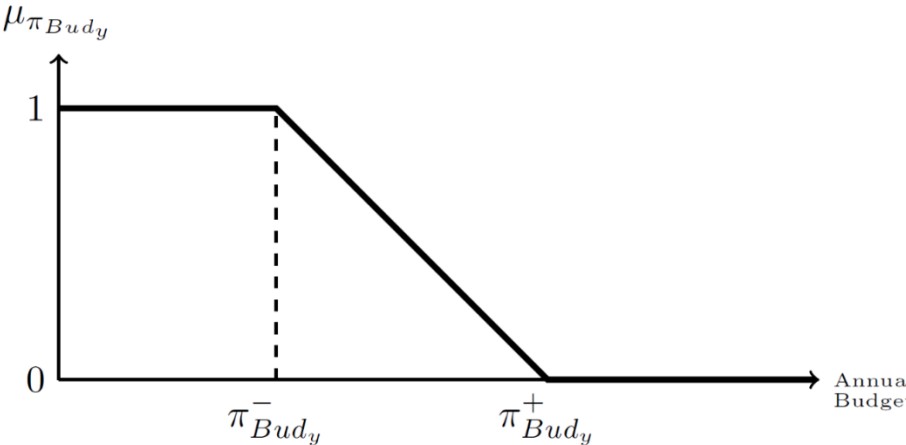

**Figure 4.** Fuzzy membership function for the budget constraint.

### 2.3. The Proposed Solution Algorithm

This section briefly discusses the details of how the proposed model in the previous section is obtained from fuzzy to the deterministic transformation of the objectives and constraints. There exists an extensive number of effective methods for reducing fuzzy linear programs in crisp systems. In order to make the best choice, different assumptions of the suggested procedures should be considered and compared with the actual decision problem to avoid inadequate modeling of the real problem, and also to decrease information costs. The maximum-minimum operator of Bellman and Zadeh that treats the objective in the same manner as the soft constraints are used in this study [43]. Figure 5 depicts the flow of the proposed solution algorithm.

The initial step in the methodology is the construction of the membership function for the fuzzy objective (shown in Figure 2), and finding the crisp/deterministic equivalent of GenCo's total profit function ($Z$) as shown in Equation (26). Two sub-models, named as pessimistic and optimistic models, have to be solved to obtain the upper/optimistic ($Z^+$) and the lower/pessimistic ($Z^-$) values of GenCo's total profit as shown in Figure 5. The optimistic sub-model uses the optimistic value for market price escalation ($EF^{MP^+}$) as an input. The objective function of the optimistic sub-model is set as the left-hand side of Equation (2), and the $\lambda$ value in constraints (3) and (4) is set as zero. The constraints between Equations (5) and (23) remain unchanged.

The optimal value of the optimistic model is $Z^+$ as shown in Figure 2. In the same manner, the pessimistic sub-model is solved by using the pessimistic value of market price escalation ($EF^{MP^-}$) and one as the value of $\lambda$ in constraints (3) and (4). After optimally solving pessimistic and optimistic sub-models, the deterministic equivalent of the objective function can be written as in Equation (26) and integrated into the model as in Equation (2). Next, crisp constraints are constructed and added to the model for hydropower generation and annual budget constraints, as explained in Equations (28) and (30), respectively. As a final step, the deterministic mixed-integer programming model between Equations (1) and (24) is solved to the optimality.

The output of the model is the optimal solution vectors of $p_{i,y,m,b}$, $p_{y,m,b}^{BIC}$, $p_{y,m,b}^{DAM}$, $\omega_{i,y}$, $u_{i,y}$, $v_{i,y}$, $x_{i,y}$, $pm_{i,y,m}$ and $\lambda$ as depicted in Figure 5. Additionally, the sensitivity analysis on the percent deviation in optimistic and pessimistic values of hydroelectric outputs of units, which occurs as a result of decreasing precipitation (due to climatic changes), is performed.

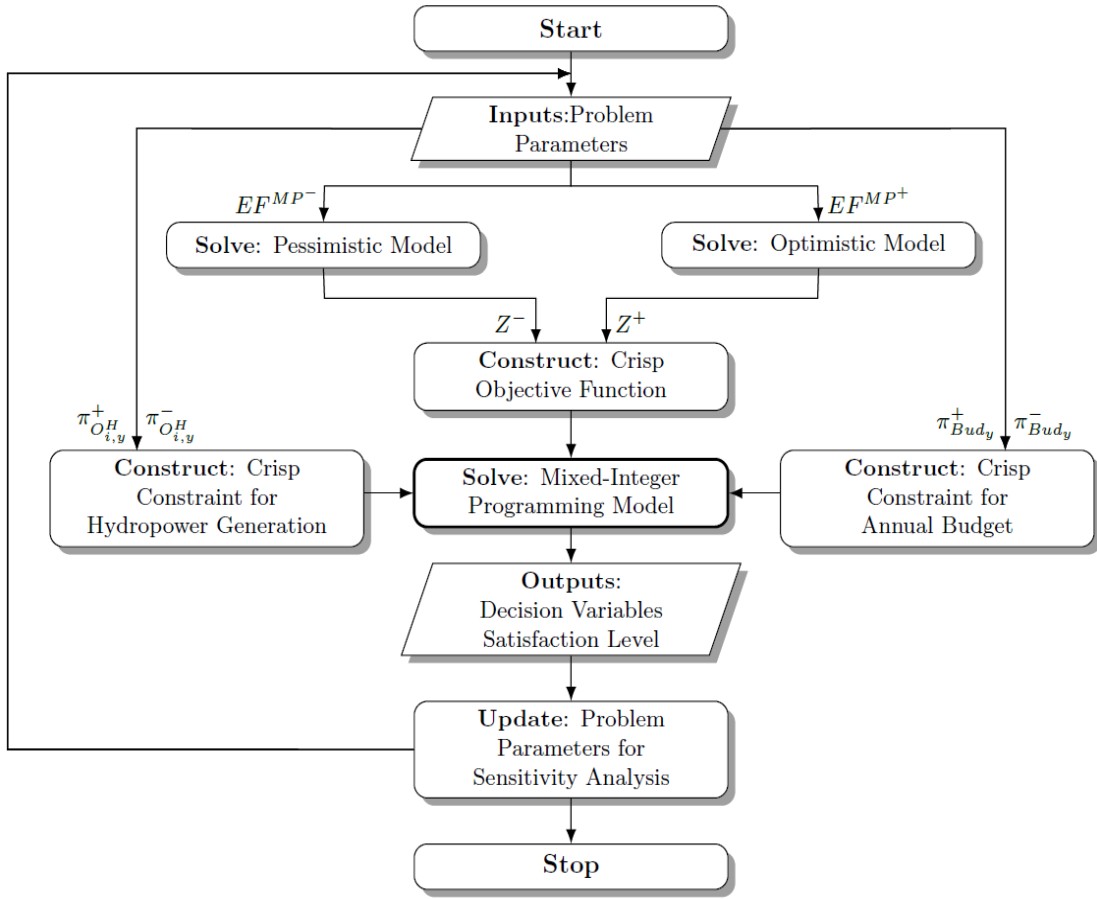

**Figure 5.** The flow chart of the solution algorithm.

### 3. The Case Study for the Proposed Model

Electrical energy producers and their long-term investment decisions cannot be independent of the current economic environment. Probability or probabilistic distribution functions become deficient in explaining uncertainties caused by unexpected crises and economic stagnations in developing countries such as Turkey. Encountered economic stagnations can affect the electricity demand, and indirectly, the profitability of GenCo. On the other hand, electricity plants are investments that take a long time to build and are partially or fully unrecoverable when market conditions have changed [44]. Therefore, an increase in the total installed capacity does not rapidly diminish along with economic stagnation. In such a fuzzy environment, the real decision maker may want to adjust the objective function to a precisely undefinable intended level rather than maximize or minimize the function. In the wake of the interviews made with the high-level management of electricity generation companies, the mixed-integer linear programming model that takes account of MTP was re-analyzed, by dropping the increase rate of electricity prices from 6% to 4%, and the obtained optimal objective function value was determined as the lowest total profit amount, to which the decision-maker would agree [45].

The optimal analysis result of the mixed-integer linear programming model that takes account of MTP with a 6 percent increase in electricity prices is established as the optimal objective function value that the GenCo targets. The nominal discount rate (3%) is applied for cash flows in the fuzzy profit target function and future fixed and variable O&M costs are determined via an escalation factor on O&M costs in the base year. Electricity prices during the planning horizon are classified as peak, intermediate, and baseload prices on a monthly basis. If the GenCo requires, it can tolerate an approximate 10% increase in the annual investment budget by using owned credit facilities.

The Turkish electricity wholesale market consists of the BICs (≈70%), DAM (≈21%), balance market (BM) (3%), and imbalance (6%). Hence, BIC and DAM are the foremost revenue basis of the GenCos that could determine the weighted average hourly Megawatt (MW) prices of their BICs for a given time. Based on the autonomous Energy Market Regulatory Authority (EMRA) in Turkey, 5/24, 11/24, and 8/24 h of a month are used as peak, intermediate, and baseload blocks, respectively. The BIC and DAM prices for every hour of the base year are given in Table 1. In order to calculate the average market-clearing prices for a given load block, hourly market clearing prices (DAM prices) announced by Energy Exchange Istanbul (EXIST) are used [46]. In order to get information about BIC prices for the base year, we interviewed the managers of GenCo [47].

**Table 1.** BIC and market-clearing prices for the base year [46,47].

| | Average BIC Prices | | | Average Market Clearing Prices | | |
|---|---|---|---|---|---|---|
| Months | Peak Load (USD/MWh) | Intermediate Load (USD/MWh) | Base Load (USD/MWh) | Peak Load (USD/MWh) | Intermediate Load (USD/MWh) | Base Load (USD/MWh) |
| 1 | 74.1 | 75.4 | 73.8 | 92.9 | 88.1 | 61.4 |
| 2 | 79.9 | 78.7 | 82.3 | 102.5 | 93.5 | 76.3 |
| 3 | 74.4 | 72.9 | 75.1 | 84.9 | 68.2 | 59.4 |
| 4 | 75.5 | 72.1 | 70.8 | 78.3 | 67.4 | 47.3 |
| 5 | 73.2 | 72.2 | 70.0 | 85.5 | 83.1 | 65.3 |
| 6 | 68.4 | 71.2 | 68.9 | 96.7 | 87.5 | 56.5 |
| 7 | 98.6 | 84.8 | 57.1 | 105.6 | 100.3 | 73.6 |
| 8 | 94.6 | 84.8 | 58.4 | 101.4 | 97.2 | 70.2 |
| 9 | 100.1 | 80.1 | 51.8 | 106.5 | 94.5 | 59.6 |
| 10 | 93.8 | 80.5 | 53.1 | 101.1 | 94.5 | 58.5 |
| 11 | 91.4 | 80.2 | 48.2 | 98.2 | 92.7 | 53.9 |
| 12 | 93.3 | 82.1 | 53.5 | 101.8 | 66.6 | 60.0 |

The GenCo expects a 6% per annual (p.a.) increase $\left(EF^{MP+} = 6\%\right)$ in wholesale prices for 6.5% of electricity demand growth. If the economy slows down the GenCo expects that electricity demand growth will decrease to 3.3% p.a. It is expected that electricity prices will increase by 4% p.a $\left(EF^{MP-} = 4\%\right)$. In addition, the GenCo desires to allocate at least 40% $\left(\alpha_y\right)$ and at most 80% $\left(\beta_y\right)$ of its load generation to BICs for every planning year.

The competition protection rule of the Turkish Electricity Market Law states that a GenCo cannot have more than 20% of the total installed capacity of the Market in the previous year ($\theta = 20\%$). The nationwide generation capacity forecasts of the Turkish Electricity Transmission Company are used for the estimated total capacity of Turkey in a given year $\left(P_y^{\text{TR}}\right)$.

The proposed long-termed investment-planning model includes both existing generation units already commissioned $\left(I^{\text{E}}\right)$ and units in commissioning plans of the GenCo $\left(I^{\text{Prj}}\right)$. Tables 2 and 3 present the data of the existing units $\left(I^{\text{E}}\right)$ and already projected units $\left(I^{\text{Prj}}\right)$, relatively [47]. The optimistic values of maximum hydroelectric outputs of the hydro units $\left(\pi^+_{O^H_{i,y}}\right)$ are calculated according to future precipitation (geo- climate) projections, the features of the basins that they are located, their turbine technologies and installed capacities.

**Table 2.** Data of the existing generation units.

| Code of the Units | Installed Capacity (MW) | $T_i^{\text{Ini}}$ (Years) | $\pi_{O_{i,y}^H}^+$ (MWh/Years) |
|---|---|---|---|
| Hydroelectric-1 | 7.00 | 12 | 28,040 |
| Hydroelectric-2 | 48.00 | 8 | 74,800 |
| Hydroelectric-3 | $1 \times 16.00$; $1 \times 14.00$ | 6 | 108,670 |
| Hydroelectric-4 | 142.00 | 1 | 359,794 |
| Hydroelectric-5 | 89.00 | 1 | 202,560 |
| Wind-1 | $13 \times 2.30$ | 1 | - |
| Wind-2 | $15 \times 2.20$ | 0 | - |
| CCCNG-1 | 120.00 | 15 | - |
| CCCNG-2 | 65.00 | 9 | - |
| ACCNG-1 | 120.00 | 10 | - |
| ACCNG-2 | 65.00 | 10 | - |
| ACCNG-3 | $1 \times 486.50$; $1 \times 450.00$ | 2 | - |
| Lignite-1 | $1 \times 150$ | 35 | - |
| Lignite-2 | $1 \times 150$ | 35 | - |

**Table 3.** Generation units' data in the commissioning plans.

| Unit Codes | Installed Capacity (MW) | $T_i^{\text{Ini}}$ (Years) | $C_i^{\text{Inv}}$ (USD/MW) | $\pi_{O_{i,y}^H}^+$ (MWh/Years) |
|---|---|---|---|---|
| Hydroelectric-6 | $1 \times 204.00$; $1 \times 3.90$ | −1 | 1,625,000 | 457,408 |
| Hydroelectric-7 | $1 \times 100.00$; $1 \times 3.20$ | −1 | 1,625,000 | 275,057 |
| Hydroelectric-8 | 8.00 | −1 | 1,625,000 | 50,860 |
| Hydroelectric-9 | $1 \times 178.89$; $1 \times 2.92$ | −2 | 1,625,000 | 741,030 |
| Hydroelectric-10 | 156.00 | −1 | 1,625,000 | 381,360 |
| Hydroelectric-11 | 20.00 | −1 | 1,625,000 | 46,657 |
| Hydroelectric-12 | 45.00 | −1 | 1,625,000 | 200,510 |
| Hydroelectric-13 | 80.00 | −3 | 1,625,000 | 301,610 |
| Hydroelectric-14 | $1 \times 250.00$; $1 \times 150.00$ | −4 | 1,625,000 | 1,350,000 |
| Hydroelectric-15 | $1 \times 225.00$; $1 \times 11.84$ | −4 | 1,625,000 | 831,980 |
| Hydroelectric-16 | $1 \times 220.00$; $1 \times 60.00$ | −4 | 1,625,000 | 919,962 |
| Hydroelectric-17 | 121.00 | −2 | 1,625,000 | 487,516 |
| Hydroelectric-18 | 62.00 | −1 | 1,625,000 | 359,794 |
| Wind-2 | $50 \times 2.20$ | −1 | 1,750,000 | - |
| Wind-3 | $11 \times 3.00$ | −1 | 1,750,000 | - |
| ACCNG-4 | $2 \times 500.00$ | −3 | 750,000 | - |
| Lignite-3 | $3 \times 150.00$ | −4 | 1,720,000 | - |

We generated a hypothetical GenCo to test our model by using data such as installed capacity, start-up date, and the expected maximum production output in public available project files of GenCos in Turkey and EMRA databases [48]. The existing generation units set of the proposed model include already commissioned old generation units in a database of EMRA and generation units in commissioning plans of the GenCo that are licensed by EMRA. Additionally, the candidate units to be added to the generation portfolio of the GenCo are named as candidate units.

Table 4 summarizes the details of the candidate units $\left(I^C\right)$ added to the GenCo's generation portfolio [48]. Additionally, Table 5 states the parameters of generation units used in the GEP model.

**Table 4.** The candidate units' data.

| | | | Candidate Units | | |
| --- | --- | --- | --- | --- | --- |
| Unit Type | Number of Units | $P^{\text{Max}}_{i,y}$ (MW) | $C^{\text{Inv}}_i$ ($10^6 \times$ USD/MW) | $T^{\text{Cons}}_i$ (Yrs) | $\pi^+_{O^H_{i,y}}$ ($10^6 \times$ MWh/Yr) |
| Hydroelectric | 10 | $1 \times 50$ | 1.625 | 4 | 150–75 |
| Wind | 60 | $2 \times 5$ | 1.750 | 3 | - |
| CCCNG | - | - | - | - | - |
| ACCNG | 10 | $1 \times 120$ | 0.750 | 3 | - |
| Lignite | 10 | $1 \times 100$ | 1.720 | 4 | - |

**Table 5.** The parameters of generation units.

| Unit Type | $C^{\text{VOM}}_{i,0}$ (USD/MWh) | $C^{\text{FOM}}_{i,0}$ (USD/MWh-Year) | $EFOR_{i,y}$ (%) | $T^{\text{L}}_i$ (Years) |
| --- | --- | --- | --- | --- |
| Hydroelectric | 2.500 | 18,000 | ~3 | 50 |
| Wind | 0 | 45,635 | ~70–80 | 25 |
| CCCNG | 73.233 | 13,170 | ~6 | 30 |
| ACCNG | 66.717 | 15,370 | ~6 | 30 |
| Lignite | 33.096 | 37,800 | ~8 | 40 |

For simplicity, a candidate wind unit contains five wind turbines having a 2 MW capacity each. It is also assumed that wind farms and hydroelectric resources with high generation efficiency are utilized firstly as a result of economic behavior.

For calculating fuel price escalation factors, aggregate coal prices of lignite mines near the locations of the lignite units and natural gas sale prices for power plants are used. The natural gas $\left(EF^{\text{NG}}_i\right)$ and lignite escalation factors $\left(EF^{\text{L}}_i\right)$ are 7.5% and 9%, respectively. In addition, it is assumed that fixed and variable O&M costs $\left(C^{\text{O\&M}}_i \text{ and } C^{\text{FOM}}_i\right)$ of generation units are increasing in the same way $\left(EF^{\text{O\&M}}_i = 3\%\right)$.

## 4. Computational Results

The 2.67 GHz Intel® Core™ i5 CPU M480 processor with 4 GB RAM, running on 64-bit Windows operating system is used to carry out all tests. GAMS 23.5 using CPLEX Academic Studio 12.0.1 is used to implement the model with 137,775 equations and 82,753 variables (including 22,980 binary variables).

According to [1], one of the two 35-year-old lignite power plants has very high improvement costs, whereas 125 K USD/MW was spent on the other's improvement. Post-improvement economic lives of generation units are increased by 8 years, and marginal costs are declined approximately by 3.60 USD/MWh.

The objective function value and the tolerance value were determined as $Z^+$ = USD 8.252 billion and $Z^+ - Z^+$ = USD 2.129 billion, respectively, with a decade planning horizon. The reserved annual budget for the new generation units and improvement of current generation units is USD 300 million with USD 30 million as the maximum allowable excess amount.

Although the deviation in maximum hydroelectric output $\left(\varphi = \left(\pi^+_{O^H_{i,y}} - \pi^+_{O^H_{i,y}}\right) / \pi^+_{O^H_{i,y}}\right)$ that hydroelectric generation units can yield was increased in 5% increment intervals between 5–75% as the fuzziness in the energy output of hydroelectric generation units increased, how energy generators substituted their hydroelectric power plant investments with various kinds of technological investments was researched. Figure 6 presents the relationship between $\varphi$, $\lambda$ and total profit. As $\varphi$ increases, the total profit decreases from its desired level (USD 8.252 billion) to the lowest acceptable amount (USD 6.123 billion).

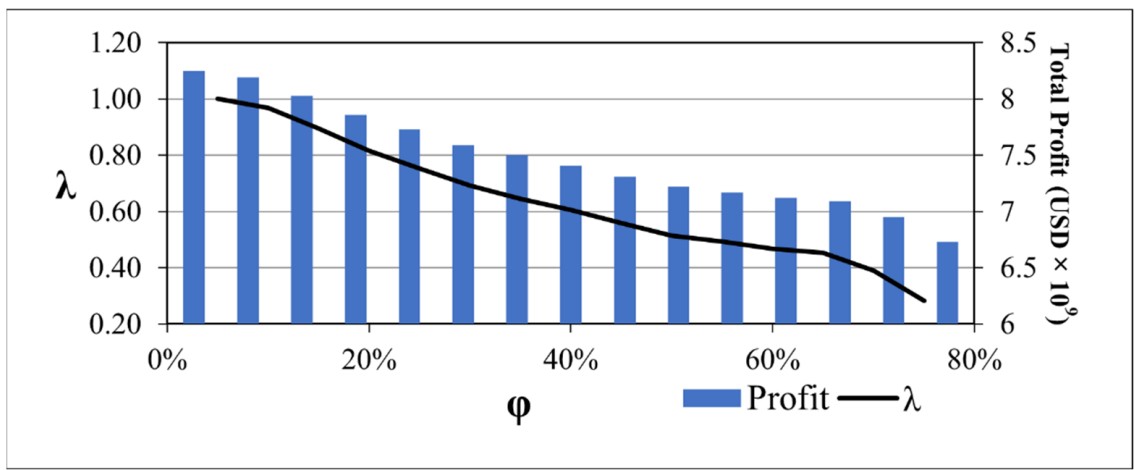

**Figure 6.** The relationship between $\varphi$, $\lambda$, and total profit.

Figure 7 and Table 6 present the fluctuation in the electric energy generation amount of the GenCo based on different $\varphi$ values from the maximum electric energy output level $\left(\pi_{O_{i,y}^H}^+\right)$ that hydroelectric generation units can reach. As $\varphi$ increases, the maximum electrical energy output that hydroelectric generation units can produce diminishes based on the technological constraint (3). Hence, the total electricity generation amount for hydroelectric generation units of the GenCo reduces.

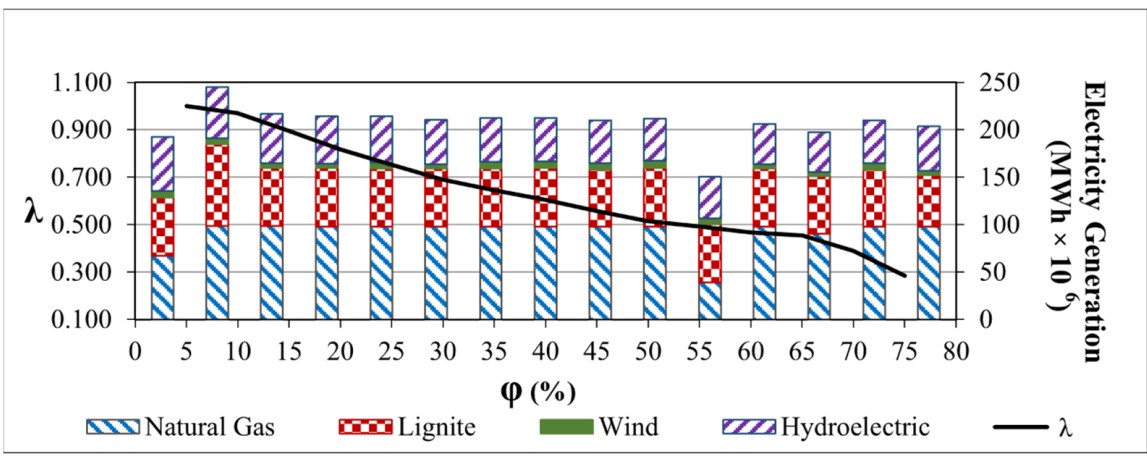

**Figure 7.** The relationship between $\varphi$, $\lambda$, and electric energy generation.

Turkey's 50-year average surface temperature increased by 1 degree and reached 14.2 degrees in 2019. Turkey's 2020 average temperature was 14.9 °C. This value is 1.4 °C above the 1981–2010 normal (13.5 °C). With this result, 2020 became the third hottest year since 1971 [49]. Hydroelectric power generation in Turkey was 88,822.8 MW in 2019. As average temperature and evaporation increased in 2020, hydroelectric power generation in Turkey decreased to 78,094.3 MW. In 2020 wind and natural gas power generation in Turkey increase, respectively, from 21,730.7 MW to 24,828.2 MW and from 57,288.2 MW to 70,931.3 MW [50]. However, lignite power generation decreased from 46,872.2 to 37,938.4 MW since COVID-19 disrupted the imported lignite coal supply chain. That is why hydroelectric power losses are compensated by more expensive natural gas power. Thus, it can be said that our research results and this real-life case are coherent.

**Table 6.** The relationship between $\varphi$, $\lambda$, and the change in electric energy generation over the years.

| Generation Unit | $\varphi$ (%) | $\lambda$ | Electricity Generation (KWh) | | | | | | | | | | |
|---|---|---|---|---|---|---|---|---|---|---|---|---|---|
| | | | 1 | 2 | 3 | 4 | 5 | 6 | 7 | 8 | 9 | 10 | Total |
| Hydroelectric | 5 | 1 | 2,384,492 | 3,546,008 | 3,819,699 | 6,725,527 | 6,726,604 | 6,740,772 | 6,735,090 | 6,730,789 | 6,757,230 | 6,748,986 | 56,915,197 |
| Wind | | | 410,549 | 410,554 | 410,570 | 733,135 | 771,940 | 809,056 | 845,463 | 882,606 | 918,178 | 937,594 | 7,129,646 |
| Natural Gas | | | 5,767,450 | 5,190,704 | 8,725,320 | 9,022,111 | 7,691,677 | 6,892,306 | 7,468,784 | 6,178,098 | 6,041,056 | 4,349,568 | 67,327,072 |
| Lignite | | | 2,185,920 | 2,185,920 | 2,185,920 | 5,464,800 | 6,557,760 | 6,557,760 | 7,650,720 | 8,743,680 | 9,836,640 | 9,836,640 | 61,205,760 |
| Hydroelectric | 10 | 0.969 | 2,237,969 | 3,320,809 | 3,590,631 | 6,312,392 | 6,340,507 | 6,344,445 | 6,343,034 | 6,314,987 | 6,340,851 | 6,343,505 | 53,489,132 |
| Wind | | | 410,549 | 410,554 | 410,570 | 736,331 | 736,401 | 773,605 | 810,666 | 829,306 | 829,346 | 829,302 | 6,776,628 |
| Natural Gas | | | 7,748,486 | 7,439,075 | 14,831,167 | 12,668,286 | 10,873,060 | 10,513,468 | 10,577,242 | 9,696,401 | 8,045,460 | 6,458,900 | 98,851,544 |
| Lignite | | | 2,185,920 | 2,185,920 | 6,312,392 | 5,464,800 | 6,557,760 | 6,557,760 | 7,650,720 | 28,136,482 | 9,836,640 | 10,929,600 | 85,817,994 |
| Hydroelectric | 15 | 0.896 | 2,203,493 | 3,266,970 | 3,528,056 | 6,213,218 | 6,213,218 | 6,213,218 | 6,213,218 | 6,213,218 | 6,213,218 | 6,213,218 | 52,491,041 |
| Wind | | | 416,270 | 416,270 | 416,270 | 417,439 | 453,214 | 490,969 | 490,969 | 492,365 | 490,969 | 490,969 | 4,575,702 |
| Natural Gas | | | 7,825,657 | 7,471,127 | 12,800,803 | 12,793,537 | 11,133,040 | 11,034,786 | 10,981,747 | 10,105,554 | 8,122,231 | 6,612,725 | 98,881,207 |
| Lignite | | | 2,185,920 | 2,185,920 | 2,185,920 | 5,464,800 | 6,557,760 | 6,557,760 | 7,650,720 | 8,743,680 | 9,836,640 | 9,836,640 | 61,205,760 |
| Hydroelectric | 20 | 0.817 | 2,094,978 | 3,113,997 | 3,364,330 | 5,908,938 | 5,931,314 | 5,931,314 | 5,931,314 | 5,908,938 | 5,931,314 | 5,931,314 | 50,047,749 |
| Wind | | | 410,576 | 410,576 | 410,576 | 410,576 | 447,793 | 484,952 | 484,952 | 484,952 | 484,952 | 484,952 | 4,514,855 |
| Natural Gas | | | 7,733,822 | 7,379,291 | 12,653,481 | 12,596,094 | 11,028,160 | 10,929,037 | 10,876,867 | 9,970,054 | 8,074,788 | 6,555,485 | 97,797,078 |
| Lignite | | | 2,185,920 | 2,185,920 | 2,185,920 | 5,464,800 | 6,557,760 | 6,557,760 | 7,650,720 | 8,743,680 | 9,836,640 | 10,929,600 | 62,298,720 |
| Hydroelectric | 25 | 0.753 | 2,031,975 | 3,021,054 | 3,264,194 | 5,729,305 | 5,748,732 | 5,748,789 | 5,748,789 | 5,729,248 | 5,748,789 | 5,748,789 | 48,519,666 |
| Wind | | | 410,576 | 410,576 | 410,576 | 717,368 | 756,147 | 794,170 | 794,170 | 794,177 | 794,170 | 794,170 | 6,676,101 |
| Natural Gas | | | 7,733,822 | 7,379,291 | 12,653,481 | 12,596,094 | 11,028,160 | 10,929,037 | 10,876,867 | 9,970,054 | 8,074,788 | 6,555,485 | 97,797,078 |
| Lignite | | | 2,185,920 | 2,185,920 | 2,185,920 | 5,464,800 | 6,557,760 | 6,557,760 | 7,650,720 | 8,743,680 | 9,836,640 | 9,836,640 | 61,205,760 |
| Hydroelectric | 30 | 0.691 | 1,984,522 | 2,950,270 | 3,187,660 | 5,592,893 | 5,610,575 | 5,610,226 | 5,610,557 | 5,592,858 | 5,610,632 | 5,610,283 | 47,360,478 |
| Wind | | | 410,576 | 410,576 | 410,576 | 410,576 | 447,793 | 484,952 | 484,952 | 484,952 | 484,952 | 484,952 | 4,514,855 |
| Natural Gas | | | 7,733,822 | 7,379,291 | 12,653,481 | 12,596,094 | 11,028,160 | 10,929,037 | 10,876,867 | 9,970,054 | 8,074,788 | 6,555,485 | 97,797,078 |
| Lignite | | | 2,185,920 | 2,185,920 | 2,185,920 | 5,464,800 | 6,557,760 | 6,557,760 | 7,650,720 | 8,743,680 | 9,836,640 | 9,836,640 | 61,205,760 |
| Hydroelectric | 35 | 0.646 | 1,938,205 | 2,881,278 | 3,112,810 | 5,458,995 | 5,475,564 | 5,475,564 | 5,475,544 | 5,458,995 | 5,475,544 | 5,475,564 | 46,228,063 |
| Wind | | | 410,576 | 410,576 | 410,576 | 729,233 | 786,589 | 842,389 | 842,396 | 842,417 | 842,461 | 842,360 | 6,959,572 |
| Natural Gas | | | 7,733,822 | 7,379,291 | 12,653,481 | 12,596,094 | 11,028,160 | 10,929,037 | 10,876,867 | 9,970,054 | 8,074,788 | 6,555,485 | 97,797,078 |
| Lignite | | | 2,185,920 | 2,185,920 | 2,185,920 | 5,464,800 | 6,557,760 | 6,557,760 | 7,650,720 | 8,743,680 | 9,836,640 | 9,836,640 | 61,205,760 |
| Hydroelectric | 40 | 0.605 | 1,898,966 | 2,823,486 | 3,049,911 | 5,346,486 | 5,362,883 | 5,363,350 | 5,362,934 | 5,346,486 | 5,362,957 | 5,362,883 | 45,280,343 |
| Wind | | | 410,576 | 410,576 | 410,576 | 728,571 | 784,270 | 838,479 | 855,449 | 855,435 | 855,457 | 855,399 | 7,004,788 |
| Natural Gas | | | 7,733,822 | 7,379,291 | 12,653,481 | 12,596,094 | 11,028,160 | 10,929,037 | 10,876,867 | 9,970,054 | 8,074,788 | 6,555,485 | 97,797,078 |
| Lignite | | | 2,185,920 | 2,185,920 | 2,185,920 | 5,464,800 | 6,557,760 | 6,557,760 | 7,650,720 | 8,743,680 | 9,836,640 | 10,929,600 | 62,298,720 |

As $\varphi$ increases, the maximum electrical energy output that wind and lignite generation units can produce diminishes drastically. The loss in total electrical energy generation resulting from hydroelectric power plants has a more negative effect on the total profit. Hence, wind and lignite power plant investments decrease compared with the total investment amount as $\varphi$ increases. Thus, the negative effect of the decrease in the amount of electrical energy output generated by hydroelectric power plants on the total profit decreases with the decrease in the total investment cost and total operation cost, which negatively affect the value of the total profit.

Figure 8 presents the wholesale electricity market activities of the GenCo for different $\varphi$ values from the maximum electric energy output level $\left( \pi_{O_{i,y}^H}^+ \right)$ that hydroelectric generation units can produce. Based on Figures 7 and 8, as $\varphi$ values increase from 5% to 10%, the resulting loss in the total electricity generation from hydroelectric power plants and the resulting drop in total profit is compensated with a lignite power plant investment, as lignite plant's equivalent factor of generation-readiness is high (equivalent forced outage rate is low). The total profit diminishes, despite the fact that the increase in the electricity generation output of lignite power plants since the marginal costs of fossil fuel power plants are relatively higher than the renewable resources power plants.

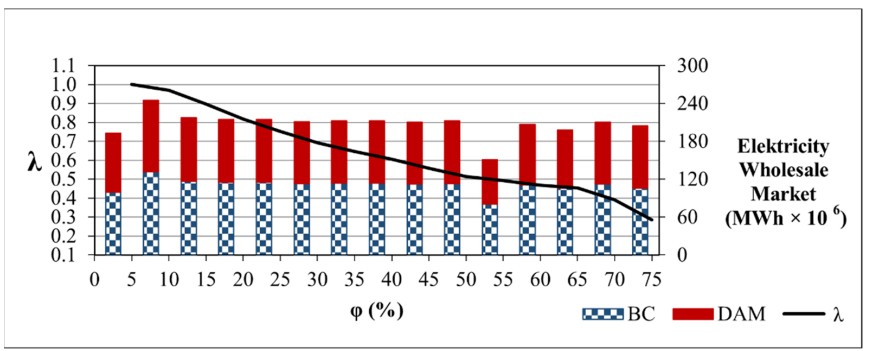

**Figure 8.** The relationship between $\varphi$, $\lambda$, and the electric energy wholesale market.

Based on Table 6, as $\varphi$ values increase from 50% to 55%, the electricity generation (sale) amount of wind and lignite power plants decreases drastically. The losses in the total electricity generation from hydroelectric power plants impact negatively on total profitability, as well as the wind turbine and lignite power plant investments diminish. Hence, the negative effects of the drop in the total investment costs and the total operation costs, and the reduction in the electric energy output level of hydroelectric power plants are reduced on total profitability.

Table 7 and Figure 9 provide the relationship between $\varphi$ and $\lambda$ values and candidate generation unit investments. Figure 10 presents the change in the total installed capacity of the GenCo based on different $\varphi$ and $\lambda$ values.

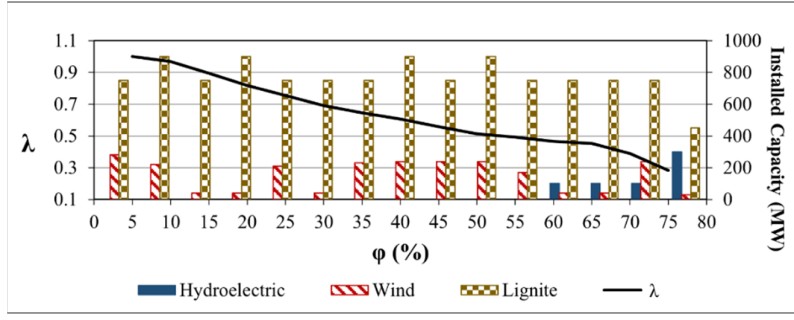

**Figure 9.** The relationship between $\varphi$, $\lambda$, and investments.

**Table 7.** The relationship between $\varphi$, $\lambda$, and the change in the investment plans.

| $\varphi$ (%) | $\lambda$ | Profit (×10⁹ USD) | Generation Type | Candidate Generation Unit Investments (MW) | | | | | | | |
|---|---|---|---|---|---|---|---|---|---|---|---|
| | | | | 4 | 5 | 6 | 7 | 8 | 9 | 10 | Total |
| 5 | 1 | 8.252 | Wind | 170 | 20 | 20 | 20 | 20 | 20 | 10 | 280 |
| | | | Lignite | | 150 | 150 | 150 | 150 | 150 | | 750 |
| 10 | 0.969 | 8.186 | Wind | 170 | | 20 | 20 | 10 | | | 220 |
| | | | Lignite | | 150 | 150 | 150 | 150 | 150 | 150 | 900 |
| 15 | 0.896 | 8.03 | Wind | | 20 | 20 | | | | | 40 |
| | | | Lignite | | 150 | 150 | 150 | 150 | 150 | | 750 |
| 20 | 0.817 | 7.862 | Wind | | 20 | 20 | | | | | 40 |
| | | | Lignite | | 150 | 150 | 150 | 150 | 150 | 150 | 900 |
| 25 | 0.753 | 7.727 | Wind | 170 | 20 | 20 | | | | | 210 |
| | | | Lignite | | 150 | 150 | 150 | 150 | 150 | | 750 |
| 30 | 0.691 | 7.5942 | Wind | | 20 | 20 | | | | | 40 |
| | | | Lignite | | 150 | 150 | 150 | 150 | 150 | | 750 |
| 35 | 0.646 | 7.4983 | Wind | 170 | 30 | 30 | | | | | 230 |
| | | | Lignite | | 150 | 150 | 150 | 150 | 150 | | 750 |
| 40 | 0.605 | 7.411 | Wind | 170 | 30 | 30 | 10 | | | | 240 |
| | | | Lignite | | 150 | 150 | 150 | 150 | 150 | 150 | 900 |
| 45 | 0.558 | 7.3113 | Wind | 180 | 30 | 30 | | | | | 240 |
| | | | Lignite | | 150 | 150 | 150 | 150 | | 150 | 750 |
| 50 | 0.514 | 7.2176 | Wind | 180 | 30 | 30 | | | | | 240 |
| | | | Lignite | | 150 | 150 | 150 | 150 | 150 | 150 | 900 |
| 55 | 0.492 | 7.1714 | Wind | | 140 | 30 | | | | | 170 |
| | | | Lignite | | 150 | 150 | 150 | 150 | 150 | | 750 |
| 60 | 0.467 | 7.1167 | Hydroelectric | | | | | | 100 | | 100 |
| | | | Wind | | 30 | 10 | | | | | 40 |
| | | | Lignite | | 150 | 150 | 150 | 150 | 150 | | 750 |
| 65 | 0.454 | 7.0893 | Hydroelectric | | | | | | 100 | | 100 |
| | | | Wind | 10 | 30 | | | | | | 40 |
| | | | Lignite | | 150 | 150 | 150 | 150 | | 150 | 750 |
| 70 | 0.39 | 6.9526 | Hydroelectric | | | | | | 100 | | 100 |
| | | | Wind | 180 | 30 | 30 | | | | | 240 |
| | | | Lignite | | 150 | 150 | 150 | 150 | | 150 | 750 |
| 75 | 0.284 | 6.7276 | Hydroelectric | | | | 100 | | 100 | 100 | 300 |
| | | | Wind | | 30 | | | | | | 30 |
| | | | Lignite | | 150 | 150 | | 150 | | | 450 |

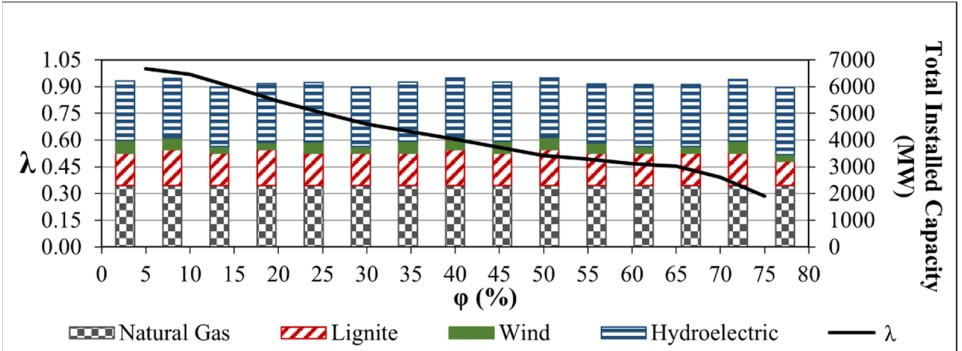

**Figure 10.** The relationship between $\varphi$, $\lambda$, and the total installed capacity at the end of the planning horizon.

As the fuzziness in $\varphi$ increases, the maximum electrical energy output that hydroelectric generation units diminish based on the technological constraint (3). As the total electricity generation amount for hydroelectric units of the GenCo and the sales income value decreases, so does the total profit (see Table 6). In order to compensate for the decline in the generation of electricity and the total profit loss, capacity investment on lignite (750 MW) and wind generation units (280 MW) are increased, whereas investments for natural gas units with their relatively high fuel costs (marginal costs) are avoided.

When $\varphi$ increases to 10%, wind turbine investments drop from 280 MW to 220 MW, and lignite generation unit investments increase from 750 MW to 900 MW. In other words, the resulting loss in the total electricity generation from hydroelectric power plants is compensated with a lignite power plant investment as the lignite plant's equivalent factor of generation readiness is high (equivalent forced outage rate is low) are expanded, whereas wind turbines investments are demoted since their equivalent factor of generation readiness is low (equivalent forced outage rate is high).

When $\varphi$ increases from 10% up to 15%, the investment in wind and lignite generation units drops to 40 MW and 750 MW, respectively, due to the drops in the income and total profit from hydroelectric generation units. In other words, investment costs that negatively affect total profit diminish.

When $\varphi$ increases from 15% up to 20%, the resulting loss in the total electricity generation from hydroelectric power plants and the resulting drop in total profit are compensated with a lignite power plant investment increase from 750 MW to 900 MW, as lignite plant's equivalent factor of generation-readiness is high (equivalent forced outage rate is low).

When $\varphi$ increases from 20% up to 25%, total lignite power plant investments drop from 900 MW to 750 MW; meanwhile, total wind power plant investments increase from 40 MW to 210 MW. In order to lessen the drop in total profit investment resources are transferred from lignite power plants with high marginal cost to wind turbines with almost zero marginal cost.

When $\varphi$ increases from 25% up to 30%, wind turbine investments, which have high unit investment costs (USD/MW) and high equivalent forced outage rate, drop from 210 MW to 40 MW. Technological investments with a low equivalent factor of generation readiness (and high equivalent forced outage rate) are decreased and the total investment cost is dropped so that the reduction in total profit caused by the loss of the hydroelectric power plant's electric energy output is compensated.

When $\varphi$ alters from 30% to 35%, the wind turbine investments with a very low marginal cost increase from 40 MW to 230 MW. When $\varphi$ increases from 35% to 40%, both the lignite power plant investments with a high equivalent factor of generation readiness are increased from 750 MW to 900 MW and the wind turbine investments with low marginal cost are increased from 230 MW to 240 MW thus the total electric energy output is increased.

When $\varphi$ increases from 40% to 45%, the lignite power plant investments with high marginal cost drop to 750 MW, both the total operating cost (variable) and the total investment costs are diminished. Hence, the drop in total profit caused by the increase in fuzziness in the maximum electric energy output that hydroelectric generation units can produce is declined. However, when $\varphi$ increases from 45% to 50%, the lignite plant investments with a high equivalent rate of generation readiness (low equivalent forced outage rate) increase back up to 900 MW so that the drop-in profitability due to the loss of electricity output of hydroelectric generation units becomes lesser. When $\varphi$ increases from 50% to 55%, both variable and fixed operating costs and the total investment costs are lessened as the lignite generation unit investments and the wind turbine investments are dropping from 900 MW to 750 MW and 240 MW to 170 MW, respectively.

Among the various values in $\varphi$ in between 5% and 55%, the decrement in the electricity output levels of hydroelectric generation units is substituted by both/either wind turbines with low marginal cost (high electricity generation cost) and/or lignite generation units with a high equivalent rate of generation readiness (low equivalent forced outage rate). However, when the uncertainty in drought or, in other words, the fuzziness in the maximum electrical

energy output level of hydroelectric generation units, is very high; the losses in electricity generation output of hydroelectric generation units are attempted to be compensated, by making more hydroelectric power plant investments. When $\varphi$ increases from 55% to 60%, the wind turbine investment drops from 170 MW to 40 MW, and a 100 MW hydroelectric generation unit investment is made. When $\varphi$ increases from 60% to 65%, no changes occur in the amount of investment; however, lignite power plant investments are postponed even though wind and hydroelectric power plant investments are scheduled in early planning periods. When $\varphi$ increases from 65% to 70%, the wind turbine investments with low electricity generation costs jump from 40 MW to 240 MW. Lastly, when $\varphi$ increases from 70% to 75%, wind turbine investments and lignite power plant investments drop from 240 MW to 30 MW and 750 MW to 450 MW, respectively, whereas the hydroelectric power plant investment increases from 100 MW to 300 MW.

## 5. Conclusions, Limitations, and Future Research Directions

This research investigates how GenCo substituted its hydroelectric power plant investments with various technological investment types when the fuzziness in the electricity generation output of hydroelectric generation units is increased. One of the contributions of the paper to the Literature is that there is no study examining long-term investment planning decisions for GenCos together with medium-term sales planning and maintenance scheduling decision problems. The existence of monthly sub-periods together with strategic investment planning decisions reduce both planning risks and affect the present value of the total profit of GenCos. It enables medium-term evaluation (monthly for each planning year) maintenance scheduling and sales planning decisions together with long-term investment planning decisions.

Another contribution is that there is almost no study in the literature about what kind of electrical energy generation technology investments can be used to substitute the hydroelectric generation unit investments in case of drought. Moreover, medium-term planning also considers seasonal and climate changes especially for the units using renewable energy sources. The results showed that wind turbines with low marginal costs and steam turbines with high energy conversion efficiency are preferable, compared with hydroelectric power plant investments when the fuzziness in hydroelectric output exists (i.e., the expectation of increasing drought conditions as a result of climate change). When the drought expectation increases and exceeds the threshold value, hydroelectric power plant investments become more preferable over steam turbines due to the high marginal cost of steam turbines even though there is low hydroelectric power potential. However, the gas turbine investments were found to be the least preferable due to high gas prices in all scenarios.

Table 6 together with Figure 7 indicate that the maximum electric energy output level of hydroelectric generation units drops as $\varphi$ increases. Hence, the total amount of generated energy for hydroelectric generation units of the GenCo is decreased. Moreover, the NPV of total sales income for hydroelectric generation units is dropped and total profit for the GenCo is reduced from its desired level (USD 8.252 billion) to the lowest acceptable amount (USD 6.123 billion) as shown in Figure 6. Finally, the total profit objective of the GenCo is diminishing, as well as $\lambda$ of hydroelectric generation units.

When the optimal investment plans of the GenCo are examined; it can be observed that until $\varphi$ reaches a threshold value (50%), the drop in the electric energy output of hydroelectric generation units can be compensated with wind turbine (low marginal cost) and lignite generation unit investments (high equivalent rate of generation readiness) as documented in both Table 7 and Figure 9. When $\varphi$ exceeds the threshold value (55%), both the lignite generation unit and wind turbine investments diminish so that NPV of variable and fixed operating costs and NPV of the total investment cost are reduced.

When the uncertainty in drought or, in other words, the fuzziness in the maximum electrical energy output level of hydroelectric generation units reaches very high values (60% and higher); the lowest profit amount acceptable by the GenCo cannot be attained

only with wind turbine and lignite generation unit investments as shown in Figure 9. Wind turbines have both high investment costs and equivalent forced outage rates, whereas lignite power plants have a low equivalent forced outage rate, but high marginal costs. Therefore, when fuzziness increases a lot in the maximum electrical energy output level that hydroelectric generation units can reach, wind turbine and lignite generation unit investments are made together with investments in hydroelectric generation units, whose investment and marginal costs are relatively lower than the first two technologies, to reach a profit level that is higher or equal to the lowest acceptable profit amount by the GenCo. However, no natural gas power plant investments are made because of extremely high marginal costs despite their low investment costs.

One of the assumptions of this GEP problem is that the number of periods operating below full capacity will increase during the year for hydroelectric power plants. According to this assumption, the maximum technically possible energy output of hydroelectric power plants in the proposed GEP problem is expressed with a fuzzy number. National and international meteorology institutes, climate change, and environmental protection organizations and institutions make predictions about the temperature, humidity, and evaporation rates for specific regions/countries in the coming years. In future studies, in order to investigate more sensitively how hydroelectric power is substituted against drought risk, the proposed GEP model can be improved by considering various scenarios developed according to these metrological forecasts.

### 5.1. Recommendations to Researchers

There is no study examining the relationship between fuel prices and electricity energy generation technology investments in the literature. Our findings show that the investment behaviors of firms with different electrical power generation technologies against different fuel prices, different electricity market structures and/or different competition conditions are potential issues to study for researchers. Moreover, future studies may offer opportunities to electricity market regulators regarding environmental regulations and renewable energy technologies incentive policies.

The results of our analysis show that the reduction in hydroelectric availability will decrease the profits of electricity producers and the drought expectation will affect their investments. Moreover, the issues need to be analyzed with an interdisciplinary approach from technical, environmental, economic, and political perspectives. Hence, researchers can develop complete and integrated models based on econometrics, finance, and operations research. For example, they can help investors by developing methods to calculate the Energy Return on Investments (EROI) of hydroelectric investments considering the effects of climate change by using various risk analyses and simulation techniques.

### 5.2. Recommendations to Regulators and Policymakers

The effects of possible environmental policies for limiting $CO_2$ emissions and imposing taxes for $CO_2$ emissions on the long-term investment plans of the power generation companies are examined with scenario and sensitivity analyses. Findings obtained from the analyses performed show that the tax policy for $CO_2$ emissions can be more effective than the policy to limit $CO_2$ emissions in terms of directing financial resources to generation unit investments using renewable resources. However, generation unit investments using renewable resources are not preferred to fossil-fueled generation unit investments without exceeding a certain threshold value for the $CO_2$ emission tax amount. In this context, energy and electricity market regulators can use the results of the research studies of micro and macro approaches to the economy, energy, and electricity markets regarding environmental policies and regulations. In addition, regulators can make more accurate and more effective decisions about environmental policies and regulations by creating their own system dynamics or energy-economy balance models.

Hydropower investments have an important aspect in the context of sustainable energy policies, but risks should not be ignored. In addition to energy and climate policies,

electricity market design and market dynamics also play an important role in hydroelectric energy investments. Hence, electricity market regulators need to start considering strategies and tools that help power generators to manage their financial risks to prevent the reduction in hydropower investments in the face of increasing hydrological and financial volatility in the future.

Regulators in the electricity market, where the energy exchange exists, can develop various financial hedging contracts and tools together with researchers for water basins that are expected to be adversely affected by climate change.

### 5.3. Recommendations for Electricity Generators

Electricity generators can adapt the proposed model to their own generation and investment portfolios, as well as include fuel contracts into the model by making the necessary modifications. They can include their fuel supply agreements in their investment models by adding constraints on the periods in which the fuel supply agreement is valid/not valid and by making the necessary modifications for the marginal cost function calculations.

The $CO_2$ emissions, service load, and cost structures of various electrical energy generation technologies are very different from each other. Therefore, electrical energy generators choose among various power plant technologies according to many criteria such as investment costs, operation and maintenance costs, power plant construction time, economic lifetime, load factor, and efficiency. In this context, electrical energy generators use one of the investment rules suitable for themselves, such as Levelized Cost of Electricity, Real Options, or Net Present Value in their investment planning models. Moreover, they can use the principle of precision equivalence to reduce risk in the proposed strategic investment planning model.

### 5.4. Limitations and Future Studies

It is assumed that there is enough demand in the market for all generated energy. The increasing rate of demand in Turkey supports this assumption. However, cycloidal and seasonal fluctuations can be encountered in the demand for electricity on a yearly and monthly basis. Moreover, due to both the long construction period of electric power plants and varying market conditions, they are partially or completely unrecoverable investments [44]. Hence, the increment rate of installed capacity will not drop as fast as the demand growth rate for electricity during economic stagnation period or seasonal fluctuations [51]. It can lead to an imbalance between supply and demand and create a demand constraint.

Probability density functions are one of the most effective tools used in expressing uncertainties in electrical energy demand. As a future study, a long-term investment planning model under the stochastic demand constraint can be considered and a stochastic programming model based on the joint change constraint problem can be developed.

It is also assumed that the independent system operator provides the necessary infrastructure to support investments in generation and distribution [52]. In reality, electrical energy generation and transmission investments may not occur simultaneously. In the long term, energy generation investments may affect the transmission system, cause regional constraints or threaten the transmission system's safety. In this context, as a future study, it is planned to develop a model with transmission constraints by using Load Flow networks in order to create optimal investment plans without ruining the security of the transmission network. Either the transmission system of Turkey or the test systems of IEEE can be used to test the model.

The results of this study show that drops in hydroelectric availability will reduce the profitability of GenCos and drought expectations affect the investments. Hence, these issues should be analyzed with a technical, environmental, economic, and political interdisciplinary approach.

**Author Contributions:** Formal analysis, B.T., F.Ç. and D.D.; investigation, H.H.T.; methodology, H.H.T., N.K., F.Ç. and D.D.; project administration, N.K.; resources, N.K.; writing—original draft, B.T.; writing—review & editing, F.Ç. and D.D. All authors have read and agreed to the published version of the manuscript.

**Funding:** This research received no external funding.

**Institutional Review Board Statement:** Not applicable.

**Informed Consent Statement:** Not applicable.

**Data Availability Statement:** Not applicable.

**Conflicts of Interest:** The authors declare no conflict of interest.

## Abbreviations

*The sets*

| | |
|---|---|
| $s(S)$ | power plants set |
| $i(I)$ | generation units set |
| $h(I^H)$ | hydroelectric generation units set |
| $m(M)$ | sub period months set |
| $b(B)$ | load levels set |
| $y(Y)$ | years set during the planning horizon |
| $I^E$ | commissioned units set or set of units in the commissioning plans, $I^E \subseteq I$ |
| $I^R$ | commissioned units set or set of units suitable for refurbishment, $I^R \subseteq I^E \subseteq I$ |
| $I^{Prj}$ | projected units set with the existing commissioning plans, $I^{Prj} \subseteq I^E \subseteq I$ |
| $I^C$ | new candidate units set selected throughout the planning horizon, $I^C \subseteq I$ |
| $I^N$ | new units set selected throughout the planning horizon, $I^N = I^{Prj} \cup I^C$, $I^N \subseteq I$ |

*Parameters*

| | |
|---|---|
| $Z^-, Z^+$ | objective function values for pessimistic and optimistic models, respectively (USD) |
| $\pi^-_{Bud_y}, \pi^+_{Bud_y}$ | pessimistic and optimistic values for the maximum budget for the refurbishment of old units and candidate unit investments at year y, respectively (USD) |
| $\pi^-_{O^H_{i,y}}, \pi^+_{O^H_{i,y}}$ | pessimistic and optimistic values for the maximum energy output of hydroelectric unit $i$ during year $y$, respectively (MWh) |
| $EF^{MP-}, EF^{MP+}$ | pessimistic and optimistic values for market price escalation, respectively (p.a.) |
| $Y$ | ending year for the planning horizon |
| $D_{y,m,b}$ | length of load block $b$ for year $y$ month $m$ (hour) |
| $DF_y$ | the discount factor for year $y$ (p.u.) |
| $DR$ | nominal discount rate (p.u.) |
| $P^{Max}_{i,y}$ | the capacity of unit $i$ for year $y$ (MW) |
| $T^L_i$ | the expected lifetime of unit $i$ (years) |
| $T^{Ini}_i$ | age of unit $i$ at the beginning of the planning horizon (years) |
| $T^{Cons}_i$ | assembly time for candidate unit $i$ (years) |
| $f(i)$ | fuel type of unit $i$, e.g., natural gas, lignite |
| $EF^{f(i)}_i$ | escalation factor for the supply cost for fuel $f$ (p.u.) |
| $EF^{CO_2}$ | escalation factor for the $CO_2$ cost (tax) (p.u.) |
| $C^{VOM}_{i,y}$ | the marginal cost for unit $i$, containing fuel cost, $CO_2$ cost, and variable part of the O&M cost through year $y$ (USD/MWh) |
| $C^{FOM}_{i,y}$ | inflation reflected fixed O&M cost for unit $i$ at year $y$ (USD/MW-year) |
| $Pr^{BIC}_{y,m,b}$ | the market price of BIC for load block $b$ at year $y$ month $m$ (USD/MWh) |
| $Pr^{DAM}_{y,m,b}$ | the market-clearing price for load block $b$ at year $y$ month $m$ (USD/MWh) |
| $HR_{i,y}$ | the heat rate for unit $i$ at year $y$ (GJ/MWh) |
| $EFOR_{i,y,m}$ | Equivalent forced outage rate for unit $i$ at year $y$ month $m$ (p.u.) |
| $\alpha_y$ | the minimum required ratio of BIC market sales to total sales (p.u.) |
| $\beta_y$ | the maximum tolerable ratio of BIC market sales to total sales (p.u.) |

| $C_i^{\text{Inv}}$ | the specific investment cost of unit $i$ (USD/MW) |
|---|---|
| $C_i^{\text{R}}$ | the specific investment cost for the refurbishment of existing unit $i$ (USD/MW) |
| $\theta$ | the maximum allowable total installed capacity of a GenCo by law based on the total installed capacity of the state in the previous year (p.u.) |
| $P_y^{\text{TR}}$ | the expected total capacity of the state in year $y$ (MW) |
| $T_i^{\text{L}_\text{R}}$ | the expected lifetime of unit $i$ after refurbishment (years) |
| *Continuous decision variables* | |
| $\lambda$ | agreed satisfaction level for fuzzy constraints and fuzzy goals, $\lambda \in [0,1]$ |
| $p_{i,y,m,b}$ | power yield of unit $i$ for load level $b$ of year $y$ month $m$ (MW) |
| $p_{y,m,b}^{\text{BIC}}$ | power sold amount in BIC market at load level $b$ of year $y$ month $m$ (MW) |
| $p_{y,m,b}^{\text{DAM}}$ | power sold amount in DAM at load level $b$ of year $y$ month $m$ (MW) |
| *Binary decision variables* | |
| $\omega_{i,y}$ | the start-up decision (commissioning) of the new unit $i$ in year $y$ |
| $u_{i,y}$ | the status of the new unit $i$ in year $y$ (1 if the unit is commissioned) |
| $x_{i,y}$ | the refurbishment decision of the old unit $i$ in year $y$ (1 if the unit is refurbished) |
| $v_{i,y}$ | the refurbishment status of the old unit $i$ in year $y$ (1 if the unit is commissioned) |
| $pm_{i,y,m}$ | the maintenance status of unit $i$ at month $m$ of year $y$ (1 if it is on maintenance) |

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
