# Peer review of "A Fuzzy Prescriptive Analytics Approach to Power Generation Capacity Planning"

_energies, doi:10.3390/en15093176_

Round 1

Reviewer 1 Report

This study investigates a power generation company's long-term power capacity investment problem (GenCo). The mid-term planning decisions such as maintenance and refurbishment scheduling of power plants are also considered in the studied investment planning problem. In the modelled electricity market, it is assumed that GenCos do business in uncertain market conditions with both bilateral contracts (BIC) and day-ahead market (DAM) transactions. The problem is modelled as a fuzzy mixed-integer linear programming model with a fuzzy objective and constraints to handle the imprecision regarding the electricity market (e.g., prices) and environmental factors (e.g., hydroelectric output due to drought). Bellman and Zadeh’s max-min criterion transforms the fuzzy capacity investment model into a model with crisp objectives and constraints.

Equations are not written in good mathematical notations.

Some are not numbered.

Many typo errors. The paper needs critical proofreading.

Tables need discussion and summarizing the most important findings.

The conclusions are very good. But I see bullet points will give more sense.

The reference style is not good.

Recent references can be used from generation capacity planning with renewables. At least from MDPI recent publications.

Input data are not clear to me. It can be summarized in an appendix.

Author Response

Thank you for your constructuve comments and invaluable suggestions. Here are our point-by-point responses.  

Equations are not written in good mathematical notations.
Thanks for this comment. We made some reformatting, but in general notation, we used subscripts and superscripts in model representation since most of the equations have multi-dimensions with separately defined sets.    

Some are not numbered.
Thank you. We have checked and numbered all of them.

Many typo errors. The paper needs critical proofreading.
Copy editing is performed

Tables need discussion and summarizing the most important findings.
We have revised the discussion and the finding for the results.

The conclusions are very good. But I see bullet points will give more sense.
We made changes to implement your suggestions. The reference style is revised based on the journal requirements. 

Recent references can be used from generation capacity planning with renewables. At least from MDPI recent publications. 
We included a few newer references from MDPI publications. 

Input data are not clear to me. It can be summarized in an appendix.
We have revised the input data descriptions and provided details. 

Reviewer 2 Report

I have reviewed the manuscript ID (energies-1654026) entitled “A Fuzzy Prescriptive Analytics Approach to Power Generation Capacity Planning”

Abstract: This study investigates a long-term power capacity investment problem of a power generation company (GenCo). The mid-term planning decisions such as maintenance and refurbishment scheduling of power plants are also considered in the studied investment planning problem. In the modeled electricity market, it is assumed that GenCos do business in uncertain market conditions with both bilateral contracts (BIC) and day-ahead market (DAM) transactions. The problem is modelled as a fuzzy mixed-integer linear programming model with a fuzzy objective and fuzzy constraints to handle the imprecisions regarding both the electricity market (e.g., prices) and environmental factors (e.g., hydroelectric output due to drought). The Bellman and Zadeh’s max-min criterion is used to transform the fuzzy capacity investment model to a model with crisp objective and constraints. The applicability of methodology is illustrated by a case study on the Turkish electric market in which GenCo tries to find the optimal power generation investment portfolio that contains five various generation technologies alternatives, namely, hydropower, wind, conventional and advanced combined-cycle natural gas, and steam (lignite) turbines. The results show that wind turbines with low marginal costs and steam turbines with high energy conversion efficiency are more preferable, compared to hydroelectric power plant investments when the fuzziness in hydroelectric output exists (i.e., the expectation of increasing drought conditions as a result of climate change). Further, the results indicate that the gas turbine investments were found to be the least preferable due to high gas prices in all scenarios.

The quality of the research work presented in the study is good, and the paper is scientifically sound, with well-modified results. I strongly recommend that it be accepted for publication.

Author Response

Thank you for your constructuve comments and invaluable suggestions. Here are our point-by-point responses.  

1. Explain in detail the reasons for the research contribution or technical impact score. What is the new contribution? 

We worked on this comment, and the new contributions of this paper are explained in the 4th and 5th paragraphs of Section 1.1. and also included here: 

“In their study [35] show that due to the slow work-pace of the coal supply chain, the losses in hydroelectric energy output during drought periods in Turkey can initially be compensated by natural gas power plants. However, later, the coal power plants increase their production. They included the utilization dimension of power plants; however, they did not consider the effects of drought expectation in investment plans as we included in this manuscript.

It was examined by [42] examines approaches from various perspectives and presents a renewed and complete survey of the optimization method implementation for the hydro scheduling solution. Similarly, it was presented by [39] a comprehensive state‐of‐the‐art survey on power generation expansion planning with renewable energy sources. According to [40], there are several ways where electric power infrastructure has contributed to climate change, and how climate change affects electric power infrastructure. The optimal Generation System Expansion Plan was studied by [44] that can satisfy the increasing electricity demand while maintaining operational elements and the stability of the energy supply. They included maintenance dimension; however, they did not consider environmental factors as we included in this manuscript.”

Also, other research and technical contributions of the paper are explained in the 5th paragraph of Section 1.2, and in the 1st and 2nd paragraphs of Section 5: 

“However, not only EFOR rate but also the precipitation and evaporation level will affect the amount of optimistic and pessimistic hydropower output. That’s why the proposed model considers not only the technological limits of hydroelectric power plants but also fuzziness in their water resources caused by drought. Using short periods in long-termed GEP modeling helps to investigate both generation amount changes of power plants to compensate for the decrease in hydropower generation during drought, and how investment plans change according to the drought expectation.

This research investigates how GenCo substituted its hydroelectric power plant investments with various technological investment types when the fuzziness in the electricity generation output of hydroelectric generation units is increased. One of the contributions of the paper to the Literature is that there is no study examining long-term investment planning decisions for GenCos together with medium-term sales planning and maintenance scheduling decision problems. The existence of monthly sub-periods together with strategic investment planning decisions reduce both planning risks and affect the present value of the total profit of GenCos. It enables medium-term evaluation (monthly for each planning year) maintenance scheduling and sales planning decisions together with long-term investment planning decisions.”

“Another contribution is that there is almost no study in the literature about what kind of electrical energy generation technology investments can be used to substitute the hydroelectric generation unit investments in case of drought. Moreover, medium-term planning also considers seasonal and climate changes especially for the units using renewable energy sources. The results showed that wind turbines with low marginal costs and steam turbines with high energy conversion efficiency are preferable, compared to hydroelectric power plant investments when the fuzziness in hydroelectric output exists (i.e., the expectation of increasing drought conditions as a result of climate change). When the drought expectation increases and exceeds the threshold value, hydroelectric power plant investments become more preferable over steam turbines due to the high marginal cost of steam turbines even though there is low hydroelectric power potential. However, the gas turbine investments were found to be the least preferable due to high gas prices in all scenarios.”

 2. Concept is not clearly explained. 

To address your comment we have made several improvements: 

The first sentence of the Abstract: “This study examines the long-term energy capacity investment problem of a power generation company (GenCo), taking into account the drought threat posed by climate change in hydropower resources in Turkey.”

In the 4th paragraph of section 1.1.: “This manuscript investigates which type of power plant investments can be substituted with hydropower investments in the long run against drought risks due to climate change, and which types of power plants meet the decrease in hydroelectric production because of drought.”

Also, Figure 1 shows which kind of management decisions are supported by the proposed GEP model, and Figure 5 explains the solution algorithm of this GEP problem. 

 3. The references within the text should be written as [1] [3, 4]

 Done.

4. What are the pros and cons of the proposed method? 

One of the assumptions of this study is that the number of periods operating below full capacity will increase during the year for hydroelectric power plants. According to this assumption, the maximum technically possible energy output of hydroelectric power plants in the proposed GEP problem is expressed with a fuzzy number.

National and international meteorology institutes, climate change research organizations, and environment protection institutions make predictions about the temperature, humidity, and evaporation rates for specific regions/countries in the coming years. In future studies, in order to investigate more sensitively how hydroelectric power is substituted against drought risk, the proposed GEP model can be improved by considering various scenarios developed according to these metrological forecasts.

5. Does it work experimental? 

Excellent comment. Here are our explanations:          

The 6th paragraph of Section 4: “Turkey's 50-year average surface temperature increased by 1 degree and reached 14.2 degrees in 2019. Turkey's 2020 average temperature was 14.9°C. This value is 1.4°C above the 1981-2010 normal (13.5°C). With this result, 2020 became the third hottest year since 1971 [49]. Hydroelectric power generation in Turkey was 88.822,8 MW in 2019. As average temperature and evaporation increased in 2020, hydroelectric power generation in Turkey decreased to 78.094,3 MW. In 2020 wind and natural gas power generation in Turkey increase, respectively, from 21.730,7 MW to 24.828,2 MW and from 57.288,2 MW to 70.931,3 MW [50]. However, lignite power generation decreased from 46.872,2 to 37.938,4 MW since COVID-19 disrupted the imported lignite coal supply chain. That's why hydroelectric power losses are compensated by more expensive natural gas power. Thus, it can be said that our research results and this real-life case are coherent.”

6. Where did you use fuzzy logic? And how did you use it? More explanations.

In the model, the objective function and some of the constraints are modeled via fuzzy logic techniques to handle the imprecisions regarding both the electricity market (e.g., prices) and environmental factors (e.g., hydroelectric output due to drought).

As we explained throughout the manuscript, we used the Bellman and Zadeh’s max-min criterion to transform the fuzzy objective function and constraints into crisp objective and constraints. The details are discussed in Subsection 2.2.

 7. In conclusions authors should clearly indicate the limits of applying proposed strategy or algorithm in point of others. Then also they should propose the future research directions. What other improvement or/and application could be done. 

One of the assumptions of this article is that the number of periods operating below full capacity will increase during the year for hydroelectric power plants. According to this assumption, the maximum technically possible energy output of hydroelectric power plants in the proposed GEP problem is expressed with a fuzzy number.

National and international meteorology institutes, climate change research organizations, and environment protection institutions make predictions about the temperature, humidity, and evaporation rates for specific regions/countries in the coming years. In future studies, in order to investigate more sensitively how hydroelectric power is substituted against drought risk, the proposed GEP model can be improved by considering various scenarios developed according to these meteorological forecasts.

8. The authors should highlight the main contributions compared to recent literature. 

New contributions of this paper are explained In the 4th and 5th paragraphs of Section 1.1.: “In their study, [35] show that due to the slow work-pace of the coal supply chain, the losses in hydroelectric energy output during drought periods in Turkey can initially be compensated by natural gas power plants. However, later, the coal power plants increase their production. They included the utilization dimension of power plants; however, they did not consider the effects of drought expectation in investment plans as we included in this manuscript.

“It was examined by [42] examines approaches from various perspectives and presents a renewed and complete survey of the optimization method implementation for the hydro scheduling solution. Similarly, it was presented by [39] a comprehensive state‐of‐the‐art survey on power generation expansion planning with renewable energy sources. According to [40], there are several ways where electric power infrastructure contributes to climate change, and how climate change affects electric power infrastructure. The optimal Generation System Expansion Plan was studied by [44] that can satisfy the increasing electricity demand while maintaining operational elements and the stability of the energy supply. They included maintenance dimension; however, they did not consider environmental factors as we included in this manuscript.”

 9. Is it possible to add experimental results to improve the quality of the paper? 

The 6th paragraph of Section 4: “Turkey's 50-year average surface temperature increased by 1 degree and reached 14.2 degrees in 2019. Turkey's 2020 average temperature was 14.9°C. This value is 1.4°C above the 1981-2010 normal (13.5°C). With this result, 2020 became the third hottest year since 1971 [49]. Hydroelectric power generation in Turkey was 88.822,8 MW in 2019. As average temperature and evaporation increased in 2020, hydroelectric power generation in Turkey decreased to 78.094,3 MW. In 2020 wind and natural gas power generation in Turkey increase, respectively, from 21.730,7 MW to 24.828,2 MW and from 57.288,2 MW to 70.931,3 MW [50]. However, lignite power generation decreased from 46.872,2 to 37.938,4 MW since COVID-19 disrupted the imported lignite coal supply chain. That's why hydroelectric power losses are compensated by more expensive natural gas power. Thus, it can be said that our research results and this real-life case are coherent.”

10. What about stability of the proposed algorithm? Please justify this point.

The proposed model is tested within the context of the case study and its outcomes are found to be rather stable. That is, the finding showed close relevance to what they expected to be in a real-world system.  

Because of the fuzzy modeling, the outcome of the optimization model may vary in each iteration, but they seem to stay within an acceptable range. The resultant suggestion of the investment type stays the same, showing the stability and robustness of the proposed model. 

11. Rewrite the references according to the journal's policy. 

Done.

12. The quality of the figure must be improved. 

The high-resolution versions of the figures are provided as a separate zip file with this revision. 

 13. How did you get the values ​​in the table? Is it experimental? Explain more. 

Tables 1, 2, 3, 4, and 5 exhibit the data used as input in the proposed GEP problem, and Tables 6 and 7 show the findings of our method.   

The 3rd paragraph in Section 3: “The BIC and DAM prices for every hour of the base year is given in Table 1. In order to calculate the average market-clearing prices for a given load block, hourly market clearing prices (DAM prices) announced by Energy Exchange Istanbul (EXIST) are used [45]. In order to get information about BIC prices for the base year, we interviewed the managers of a GenCo [46].”

Tables 2 and 3 present the data of the existing units  and already projected units , relatively.

7 th paragraph of Section 3: “We generated a hypothetical GenCo to test our model by using data such as installed capacity, start-up date and the expected maximum production output in public available project files of GenCos in Turkey and EMRA databases [47]. The existing generation units set of the proposed model include already commissioned old generation units in database of EMRA and generation units in commissioning plans of the GenCo that are licensed by EMRA. Also, the candidate units to be added to the generation portfolio of the GenCo are named as candidate units.”

Table 4 summarizes the details of the candidate units  added to the GenCo’s generation portfolio [48]. Also, Table 5 states the parameters of generation units used in the GEP model. (8th paragraph of Section 3).

Note: some of the notations did not showup in this text input. So, pelase refer to the revised paper for a more complete represenattion. 

14. What rules did you use for fuzzy logic? 

The developed long-term (strategic) investment-planning model considers the fuzziness in the hydroelectric output and annual budget. The developed fuzzy mixed-integer linear programming model is solved by using the Zimmermann approach (Zimmermann, 1991; Zimmermann, 1978). We assumed linear fuzzy membership functions as in Figures 2 and 3. The details are discussed in Subsections 2.2 and 2.3. 

References

Zimmermann, H. J. (1978). Fuzzy programming and linear programming with several objective functions, Fuzzy Sets and Systems, 1 (1), 45-55.

Zimmermann, H. J. (1991). Fuzzy Set Theory and Its Applications. (2. edition), New York: Springer Science +Business Media, LLC, ISBN:978-94-015-7951-3.

15. What kind of fuzzy logic is used? Mention its features? HASAN

We assumed linear fuzzy membership functions as in Figures 2 and 3. The details are discussed in Subsections 2.2 and 2.3. Further, the properties of the approach are explained in the revised manuscript in Subsections 2.1 and Subsection 2.3. 

Reviewer 3 Report

After studying the article, there are many comments about this work, including: 1. Explain in detail the reasons for the research contribution or technical impact score. What is the new contribution.
2. concept is not clearly expalined.
3. The references within the text should be written as [1] [3, 4].... 4. What are the pros and cons of the proposed method? 5. Does it work experimental? 6. Where did you use fuzzy logic? And how did you use it? More explanations.
7. In conclusions authors should clearly indicate the limits of applying proposed strategy or algorithm in point of others. Then also they should propose the future research directions. What other improvement or/and application could be done.
8.The authors should highlight the main contributions compared to recent literature.
9. Is it possible to add experimental results to improve the quality of the paper?.
10. What about stability of the proposed algorithm? Please justify this point.
11.Rewrite the references according to the journal's policy.
12.The quality of figure must be improved.
13.
How did you get the values ​​in the table? Is it experimental? Explain more.
14.
What rules did you use for fuzzy logic?
15.What kind of fuzzy logic used? Mention its features?

Author Response

Thank you for your constructuve comments and invaluable suggestions. Here are our point-by-point responses.

1. Explain in detail the reasons for the research contribution or technical impact score. What is the new contribution?

We worked on this comment, and the new contributions of this paper are explained in the 4th and 5th paragraphs of Section 1.1. and also included here:

“In their study [35] show that due to the slow work-pace of the coal supply chain, the losses in hydroelectric energy output during drought periods in Turkey can initially be compensated by natural gas power plants. However, later, the coal power plants increase their production. They included the utilization dimension of power plants; however, they did not consider the effects of drought expectation in investment plans as we included in this manuscript.

It was examined by [42] examines approaches from various perspectives and presents a renewed and complete survey of the optimization method implementation for the hydro scheduling solution. Similarly, it was presented by [39] a comprehensive state‐of‐the‐art survey on power generation expansion planning with renewable energy sources. According to [40], there are several ways where electric power infrastructure has contributed to climate change, and how climate change affects electric power infrastructure. The optimal Generation System Expansion Plan was studied by [44] that can satisfy the increasing electricity demand while maintaining operational elements and the stability of the energy supply. They included maintenance dimension; however, they did not consider environmental factors as we included in this manuscript.”

Also, other research and technical contributions of the paper are explained in the 5th paragraph of Section 1.2, and in the 1st and 2nd paragraphs of Section 5:

“However, not only EFOR rate but also the precipitation and evaporation level will affect the amount of optimistic and pessimistic hydropower output. That’s why the proposed model considers not only the technological limits of hydroelectric power plants but also fuzziness in their water resources caused by drought. Using short periods in long-termed GEP modeling helps to investigate both generation amount changes of power plants to compensate for the decrease in hydropower generation during drought, and how investment plans change according to the drought expectation.

This research investigates how GenCo substituted its hydroelectric power plant investments with various technological investment types when the fuzziness in the electricity generation output of hydroelectric generation units is increased. One of the contributions of the paper to the Literature is that there is no study examining long-term investment planning decisions for GenCos together with medium-term sales planning and maintenance scheduling decision problems. The existence of monthly sub-periods together with strategic investment planning decisions reduce both planning risks and affect the present value of the total profit of GenCos. It enables medium-term evaluation (monthly for each planning year) maintenance scheduling and sales planning decisions together with long-term investment planning decisions.”

“Another contribution is that there is almost no study in the literature about what kind of electrical energy generation technology investments can be used to substitute the hydroelectric generation unit investments in case of drought. Moreover, medium-term planning also considers seasonal and climate changes especially for the units using renewable energy sources. The results showed that wind turbines with low marginal costs and steam turbines with high energy conversion efficiency are preferable, compared to hydroelectric power plant investments when the fuzziness in hydroelectric output exists (i.e., the expectation of increasing drought conditions as a result of climate change). When the drought expectation increases and exceeds the threshold value, hydroelectric power plant investments become more preferable over steam turbines due to the high marginal cost of steam turbines even though there is low hydroelectric power potential. However, the gas turbine investments were found to be the least preferable due to high gas prices in all scenarios.”

2. Concept is not clearly explained.

To address your comment we have made several improvements:

The first sentence of the Abstract: “This study examines the long-term energy capacity investment problem of a power generation company (GenCo), taking into account the drought threat posed by climate change in hydropower resources in Turkey.”

In the 4th paragraph of section 1.1.: “This manuscript investigates which type of power plant investments can be substituted with hydropower investments in the long run against drought risks due to climate change, and which types of power plants meet the decrease in hydroelectric production because of drought.”

Also, Figure 1 shows which kind of management decisions are supported by the proposed GEP model, and Figure 5 explains the solution algorithm of this GEP problem.

3. The references within the text should be written as [1] [3, 4]

Done.

4. What are the pros and cons of the proposed method?

One of the assumptions of this study is that the number of periods operating below full capacity will increase during the year for hydroelectric power plants. According to this assumption, the maximum technically possible energy output of hydroelectric power plants in the proposed GEP problem is expressed with a fuzzy number.

National and international meteorology institutes, climate change research organizations, and environment protection institutions make predictions about the temperature, humidity, and evaporation rates for specific regions/countries in the coming years. In future studies, in order to investigate more sensitively how hydroelectric power is substituted against drought risk, the proposed GEP model can be improved by considering various scenarios developed according to these metrological forecasts.

5. Does it work experimental?

Excellent comment. Here are our explanations:

The 6th paragraph of Section 4: “Turkey's 50-year average surface temperature increased by 1 degree and reached 14.2 degrees in 2019. Turkey's 2020 average temperature was 14.9°C. This value is 1.4°C above the 1981-2010 normal (13.5°C). With this result, 2020 became the third hottest year since 1971 [49]. Hydroelectric power generation in Turkey was 88.822,8 MW in 2019. As average temperature and evaporation increased in 2020, hydroelectric power generation in Turkey decreased to 78.094,3 MW. In 2020 wind and natural gas power generation in Turkey increase, respectively, from 21.730,7 MW to 24.828,2 MW and from 57.288,2 MW to 70.931,3 MW [50]. However, lignite power generation decreased from 46.872,2 to 37.938,4 MW since COVID-19 disrupted the imported lignite coal supply chain. That's why hydroelectric power losses are compensated by more expensive natural gas power. Thus, it can be said that our research results and this real-life case are coherent.”

6. Where did you use fuzzy logic? And how did you use it? More explanations.

In the model, the objective function and some of the constraints are modeled via fuzzy logic techniques to handle the imprecisions regarding both the electricity market (e.g., prices) and environmental factors (e.g., hydroelectric output due to drought).

As we explained throughout the manuscript, we used the Bellman and Zadeh’s max-min criterion to transform the fuzzy objective function and constraints into crisp objective and constraints. The details are discussed in Subsection 2.2.

7. In conclusions authors should clearly indicate the limits of applying proposed strategy or algorithm in point of others. Then also they should propose the future research directions. What other improvement or/and application could be done.

One of the assumptions of this article is that the number of periods operating below full capacity will increase during the year for hydroelectric power plants. According to this assumption, the maximum technically possible energy output of hydroelectric power plants in the proposed GEP problem is expressed with a fuzzy number.

National and international meteorology institutes, climate change research organizations, and environment protection institutions make predictions about the temperature, humidity, and evaporation rates for specific regions/countries in the coming years. In future studies, in order to investigate more sensitively how hydroelectric power is substituted against drought risk, the proposed GEP model can be improved by considering various scenarios developed according to these meteorological forecasts.

8. The authors should highlight the main contributions compared to recent literature.

New contributions of this paper are explained In the 4th and 5th paragraphs of Section 1.1.: “In their study, [35] show that due to the slow work-pace of the coal supply chain, the losses in hydroelectric energy output during drought periods in Turkey can initially be compensated by natural gas power plants. However, later, the coal power plants increase their production. They included the utilization dimension of power plants; however, they did not consider the effects of drought expectation in investment plans as we included in this manuscript.

“It was examined by [42] examines approaches from various perspectives and presents a renewed and complete survey of the optimization method implementation for the hydro scheduling solution. Similarly, it was presented by [39] a comprehensive state‐of‐the‐art survey on power generation expansion planning with renewable energy sources. According to [40], there are several ways where electric power infrastructure contributes to climate change, and how climate change affects electric power infrastructure. The optimal Generation System Expansion Plan was studied by [44] that can satisfy the increasing electricity demand while maintaining operational elements and the stability of the energy supply. They included maintenance dimension; however, they did not consider environmental factors as we included in this manuscript.”

9. Is it possible to add experimental results to improve the quality of the paper?

The 6th paragraph of Section 4: “Turkey's 50-year average surface temperature increased by 1 degree and reached 14.2 degrees in 2019. Turkey's 2020 average temperature was 14.9°C. This value is 1.4°C above the 1981-2010 normal (13.5°C). With this result, 2020 became the third hottest year since 1971 [49]. Hydroelectric power generation in Turkey was 88.822,8 MW in 2019. As average temperature and evaporation increased in 2020, hydroelectric power generation in Turkey decreased to 78.094,3 MW. In 2020 wind and natural gas power generation in Turkey increase, respectively, from 21.730,7 MW to 24.828,2 MW and from 57.288,2 MW to 70.931,3 MW [50]. However, lignite power generation decreased from 46.872,2 to 37.938,4 MW since COVID-19 disrupted the imported lignite coal supply chain. That's why hydroelectric power losses are compensated by more expensive natural gas power. Thus, it can be said that our research results and this real-life case are coherent.”

10. What about stability of the proposed algorithm? Please justify this point.

The proposed model is tested within the context of the case study and its outcomes are found to be rather stable. That is, the finding showed close relevance to what they expected to be in a real-world system.

Because of the fuzzy modeling, the outcome of the optimization model may vary in each iteration, but they seem to stay within an acceptable range. The resultant suggestion of the investment type stays the same, showing the stability and robustness of the proposed model.

11. Rewrite the references according to the journal's policy.

Done.

12. The quality of the figure must be improved.

The high-resolution versions of the figures are provided as a separate zip file with this revision.

13. How did you get the values ​​in the table? Is it experimental? Explain more.

Tables 1, 2, 3, 4, and 5 exhibit the data used as input in the proposed GEP problem, and Tables 6 and 7 show the findings of our method.

The 3rd paragraph in Section 3: “The BIC and DAM prices for every hour of the base year is given in Table 1. In order to calculate the average market-clearing prices for a given load block, hourly market clearing prices (DAM prices) announced by Energy Exchange Istanbul (EXIST) are used [45]. In order to get information about BIC prices for the base year, we interviewed the managers of a GenCo [46].”

Tables 2 and 3 present the data of the existing units and already projected units , relatively.

7 th paragraph of Section 3: “We generated a hypothetical GenCo to test our model by using data such as installed capacity, start-up date and the expected maximum production output in public available project files of GenCos in Turkey and EMRA databases [47]. The existing generation units set of the proposed model include already commissioned old generation units in database of EMRA and generation units in commissioning plans of the GenCo that are licensed by EMRA. Also, the candidate units to be added to the generation portfolio of the GenCo are named as candidate units.”

Table 4 summarizes the details of the candidate units added to the GenCo’s generation portfolio [48]. Also, Table 5 states the parameters of generation units used in the GEP model. (8th paragraph of Section 3).

Note: some of the notations did not showup in this text input. So, pelase refer to the revised paper for a more complete represenattion.

14. What rules did you use for fuzzy logic?

The developed long-term (strategic) investment-planning model considers the fuzziness in the hydroelectric output and annual budget. The developed fuzzy mixed-integer linear programming model is solved by using the Zimmermann approach (Zimmermann, 1991; Zimmermann, 1978). We assumed linear fuzzy membership functions as in Figures 2 and 3. The details are discussed in Subsections 2.2 and 2.3.

References

Zimmermann, H. J. (1978). Fuzzy programming and linear programming with several objective functions, Fuzzy Sets and Systems, 1 (1), 45-55.

Zimmermann, H. J. (1991). Fuzzy Set Theory and Its Applications. (2. edition), New York: Springer Science +Business Media, LLC, ISBN:978-94-015-7951-3.

15. What kind of fuzzy logic is used? Mention its features?

We assumed linear fuzzy membership functions as in Figures 2 and 3. The details are discussed in Subsections 2.2 and 2.3. Further, the properties of the approach are explained in the revised manuscript in Subsections 2.1 and Subsection 2.3.

Reviewer 4 Report

SUMMARY
The paper is complete research work in the field of power generation capacity planning by fuzzy logic modeling. However, the structure of the paper and the abundance of formulas and abbreviations make the text difficult to read and understand. A little more textual explanation should be added since the audience of the Energies is not mathematicians, for whom the language of formulas is the main one.

COMMENTS
1. Please provide quantitative data about the climate change problem in Section 1.1 "Climate change" to justify the necessity of this factor in the proposed method. Please clarify the conclusion "Thus, it is predicted that Turkey's water sources will be affected by global climate change."

2. The literature review provides a long list of the GEP studies and solution methods (lines 60-85, 145-154). However, it remains unclear why the development of new methods is required. An audience that is not an expert in the field will hardly understand it. Perhaps a brief explanation of the existing methods is needed. 

3. To understand complex mathematical expressions, a reader first wants to understand the general idea and main steps. Without it, the equations presented in the paper are difficult to understand. Please, try to explain the current constraints and functions as a block diagram. First, describe the method with text and block diagrams, and only then give formulas.

4. While reading, it is inconvenient to navigate in appendix 1; it has many notations. Perhaps it is better to insert a detailed explanation after each formula.

5. Please use the full description of the values in the text and graphs instead of their letter equivalent. Otherwise, a reader constantly has to spend time looking for notations in Appendix 1.

Author Response

Thank you for your constructive comments and invaluable suggestions. What follows are our point-by-point responses to your comments and suggestions.

1. Please provide quantitative data about the climate change problem in Section 1.1 "Climate change" to justify the necessity of this factor in the proposed method. Please clarify the conclusion "Thus, it is predicted that Turkey's water sources will be affected by global climate change."
Thank you for this comment. Since the purpose of this study was not to show climate change, per se, we used the extant literature to signify the importance of climate change as it related to the proposed research model. We hope that this approach is acceptable to the reviewer. 

2. The literature review provides a long list of the GEP studies and solution methods (lines 60-85, 145-154). However, it remains unclear why the development of new methods is required. An audience that is not an expert in the field will hardly understand it. Perhaps a brief explanation of the existing methods is needed. 
Thank you for this comment and the opportunity to clarify. In the revised manuscript, the contribution of this study is presented on lines 176-183 as “... the proposed model is novel by considering the operations of the GenCo in BIC market and DAM are integrated by using short (monthly) periods in long-termed (strategic) GEP modeling. Thus, both the mid-termed sale and power generation strategy of a GenCo are deliberated in strategic investment planning. Moreover, mid-term decisions such as maintenance scheduling of the generation units are considered in a decentralized long-termed GEP model. Also, different from [1], [22], and [42], this study used monthly equivalent forced outage rate modeling (EFOR) for each planning year in order to consider seasonal changes in the availability of generation units, especially units using renewable energy sources”.

3. To understand complex mathematical expressions, a reader first wants to understand the general idea and main steps. Without it, the equations presented in the paper are difficult to understand. Please, try to explain the current constraints and functions as a block diagram. First, describe the method with text and block diagrams, and only then give formulas.
Although we have not provided a block diagram, in the revised manuscript, we captured the proposed flow of the algorithmic logic in Figure 5. We hope that this would be, at least partially, acceptable to the reviewer. 

4. While reading, it is inconvenient to navigate in appendix 1; it has many notations. Perhaps it is better to insert a detailed explanation after each formula.
Thank you for this suggestion. We tried to do that but the paper has become too busy and convoluted. Instead, in the revised manuscript, we included several short explanations within the Appendix for further clarification. Within the text, where appropriate, each constraint is also followed with short explanations. 

5. Please use the full description of the values in the text and graphs instead of their letter equivalent. Otherwise, a reader constantly has to spend time looking for notations in Appendix 1.
You are right on this comment. It would take some time and effort to go back and forth to Appendix 1. Including notations after each and every text and graph may also further extend and crowd the paper. We tried our best to find a balance between these two points; while keeping the paper streamlined we also added necessary explanations to minimize the need to look at Appendix 1.  

Round 2

Reviewer 1 Report

The reviewer is satisfied with the corrections made. 

The manuscript can be accepted. 

Author Response

Comment: The reviewer is satisfied with the corrections made. The manuscript can be accepted. 
Response: Thank you very much for your insightful suggestions. 

Reviewer 3 Report

Hi

Dear Author,

After studying the paper, the notes are as follows:

1.Rewrite the reference according to the journal's policy.

Example of writing a reference:

  1. Nicola, M.; Nicola, C.-I. Sensorless Fractional Order Control of PMSM Based on Synergetic and Sliding Mode Controllers. Electronics 2020, 9, 1494. https://doi.org/10.3390/electronics9091494.
  2. Fukami, T.; Momiyama, M.; Shima, K.; Hanaoka, R.; Takata, S. Steady-State Analysis of a Dual-Winding Reluctance Generator With a Multiple-Barrier Rotor. in IEEE Transactions on Energy Conversion, 2008, 23(2), 492-498, doi: 10.1109/TEC.2008.918656.

2.The Appendix is always before the References.

Thanks

Author Response

Comment 1.Rewrite the reference according to the journal's policy.

Example of writing a reference:

  1. Nicola, M.; Nicola, C.-I. Sensorless Fractional Order Control of PMSM Based on Synergetic and Sliding Mode Controllers. Electronics 2020, 9, 1494. https://doi.org/10.3390/electronics9091494.
  2. Fukami, T.; Momiyama, M.; Shima, K.; Hanaoka, R.; Takata, S. Steady-State Analysis of a Dual-Winding Reluctance Generator With a Multiple-Barrier Rotor. in IEEE Transactions on Energy Conversion, 2008, 23(2), 492-498, doi: 10.1109/TEC.2008.918656.

Response: The reference style is revised and corrected based on the journal requirements.

Comment 2.The Appendix is always before the References.

Response: Thank you, the Appendix is moved to end of the text before the References section. 

Thanks

Reviewer 4 Report

The authors did not fully take into account the comments. Therefore, explanations are given below in accordance with the comments of the initial review.

1. By the "quantitative data about the climate change problem" was meant precise data about future water flow decreasing. It will not violate the purpose of your study if you extend lines 87-90 by information from [24]. For example the decreasing trend of surface waters:

"Simulation results of the water budget model demonstrated that nearly 20% of the surface waters in the studied basins would be reduced by the year 2030. This will be 35% in 2050 and 50% in 2100. The decrease in surface water will cause water scarcity in agricultural, domestic and industrial use. Besides, increased losses through evapotranspiration of plants (10% by 2030, 20% by 2050) will dramatically increase the need for irrigation water."

2. Answer accepted.

3. The main idea of this comment was implementation the top-down approach:
- describe the model general idea with text
- give the algorithm block diagram
- give the mathematical details at the end. 
The revised article still starts the model explanation from "Mathematical formulation".

4. Answer accepted.

5. Answer accepted.

Author Response

Comment 1. By the "quantitative data about the climate change problem" was meant precise data about future water flow decreasing. It will not violate the purpose of your study if you extend lines 87-90 by information from [24]. For example the decreasing trend of surface waters:

"Simulation results of the water budget model demonstrated that nearly 20% of the surface waters in the studied basins would be reduced by the year 2030. This will be 35% in 2050 and 50% in 2100. The decrease in surface water will cause water scarcity in agricultural, domestic and industrial use. Besides, increased losses through evapotranspiration of plants (10% by 2030, 20% by 2050) will dramatically increase the need for irrigation water."

Response 1. Thank you for this comment. As you suggested, in the revised manuscript we included numerical estimated calculated/suggested by [24], as follows (see lines 90-95 in the revised manuscript):

Using a simulation-based study, Aktas [24] demonstrated that nearly 20% of the surface water in the studied basins in Turkey will be vanished by the year 2030, and the trend for this loss in water reserve will increase to 35% in 2050 and 50% in 2100. Furthermore, the same study also suggests that the evapotranspiration of plants will contribute to the water loss by as much as another 20%.  

Comment 2. Answer accepted.

Response: Thank you. 

Comemnt 3. The main idea of this comment was implementation the top-down approach:
- describe the model general idea with text
- give the algorithm block diagram
- give the mathematical details at the end. 
The revised article still starts the model explanation from "Mathematical formulation".

Response: In the revised manuscript, we start the modeling section with a concise description of the motivation and the general idea of the underlying overall model (see lines 205-218 in the revised manuscript). We then provide the details about the mathematical formulation. We provide a high-level diagram block diagram later in the section.   

Comment 4. Answer accepted.

Response: Thank you. 

Comment 5. Answer accepted.

Response: Thank you. 

Round 3

Reviewer 4 Report

The article can be accepted, taking into account the fact that the authors promise: "We then provide the details about the mathematical formulation. We provide a high-level diagram block diagram later in the section."